

**Changes in Aerosol/Gas-Phase Distribution Ratio of Semi-Volatile Products**
**Affect Secondary Organic Aerosol Formation with NOx from α-Pinene**
**Photooxidation**
**Shijie Liu[1,2], Xinbei Xu[1], Si Zhang[1], Rongjie Li[1], Zheng Li[1], Can Wu[1], Rui Li[1],**
**Guiqin Zhang[2], and Gehui Wang[1,3*]**
[1]Key Laboratory of Geographic Information Science of the Ministry of Education,
School of Geographic Sciences, East China Normal University, Shanghai 200241,
China
[2]Resources and Environment Innovation Institute, Shandong Jianzhu University,
Jinan 250101, China
[3]Institute of Eco-Chongming, Cuiniao Road, Chenjia Zhen, Chongming, Shanghai
202150, China

*Correspondence to: Prof. Gehui Wang (ghwang@geo.ecnu.edu.cn)





## Abstract

Atmospheric α-pinene is one of the most important precursors of secondary organic aerosols (SOA). The formation of α-pinene derived SOA is strongly affected by NOx. However, we still do not comprehensively understand the effects of $NO_x$ on α-pinene derived SOA formation. Therefore, we conducted α-pinene photooxidation experiments in an atmospheric chamber at different $NO_x$ concentrations. The yields of α-pinene SOA increased with $NO_x$ concentration under low-$NO_x$ conditions, but were suppressed under high-NOx conditions. The maximum SOA yields were 8.0% and 26.2% in the low- and high-volatility organic compound (VOC) experiments, respectively. We found the increased SOA yields under low-NOx conditions were related to increased consumption of α-pinene. The products of α-pinene photooxidation were mainly semi-volatile, and the change in the aerosol/gas-phase distribution ratio as the formation of α-pinene photooxidation products increased was identified as the main reason for the enhanced SOA yields with increasing NOx. The sensitivity of the SOA yield to changes in NOx and VOCs under different experimental conditions was also analyzed. This study also quantified the nitrogen-containing organic compound (NOC) concentrations. The mass fraction of NOCs in SOA increased monotonically with NOx in the α-pinene photooxidation process, and the maximum NOC mass fraction made up as much as two-fifths of the α-pinene SOA.



## 1. Introduction

Fine particulate matter (PM$_{2.5}$) is an important atmospheric pollutant. Recently, PM$_{2.5}$ has attracted attention because of its negative impacts on human health, air quality, atmospheric chemical reactions, and even climate radiation balances (Bellouin et al., 2020; Wang et al., 2016). Secondary organic aerosols (SOA), which form via the atmospheric photooxidation of volatile organic compounds (VOCs), are some of the most significant sources of PM$_{2.5}$ (Lv et al., 2022; Mcfiggans et al., 2019; Zhang et al., 2018). However, due to the diversity of SOA production processes and influencing factors, much researches are still needed for the fully understand the formation mechanisms of SOA.

NO$_x$, which is mainly emitted from human activities, is a key substance that facilitates photooxidation. Differences in concentrations of NO$_x$ can lead to significant changes in photooxidation processes and further affect SOA formation (Sarrafzadeh et al., 2016; Nussbaumer et al., 2022). Organic peroxyl radicals (RO$_2$) are the main intermediate oxidation product when VOCs are oxidized by OH. The interaction between NO$_x$ and RO$_2$ plays a key role in the SOA formation process, and nonlinear relationships between NO$_x$ and SOA yields have been observed in a large number of experimental investigations (Xu et al., 2021; Zhao et al., 2018; Sarrafzadeh et al., 2016; Xu et al., 2014). Hydroperoxides (ROOH), which are formed from the reactions of RO$_2$ with HO$_2$ or RO$_2$, have low volatilities and are responsible for SOA formation. However, as the concentration of NOx increases, RO$_2$ reacts with NO and is rapidly converted to alkoxy radicals (RO). RO act as an intermediate and goes on to produce more high volatility compounds, which suppresses SOA formation (Sarrafzadeh et al., 2016; Atkinson, 2000). Highly oxygenated organic molecules (HOMs) formed by the autoxidation of RO$_2$ are also key compounds involved in SOA formation (Rissanen, 2021). HOM formation is suppressed when the NO$_2$ concentration increases and RO$_2$ + NO becomes the dominant sink of RO$_2$ in the photooxidation process, which also contributes to the inhibition of SOA formation as NO$_x$ concentrations increase.



Clearly, the effects of NOₓ on photooxidation and SOA formation are quite
complicated, but they have been widely studied in controlled chamber experiments.
Based on the semi-volatile partitioning theory in SOA formation, it has been established
that SOA yield is a function of SOA mass concentration when other experimental
conditions are held constant (Odum et al., 1996; Takeuchi et al., 2022). Furthermore,
the mass concentration of formed SOA is directly related to the mass concentration of
available VOC precursors. However, the SOA yield is often discussed as a constant,
and the nonlinear relationships between SOA yield and initial NOx concentration
reported in chamber studies do not account for the consumption of VOCs (Chen et al.,
2022b; Aruffo et al., 2022; Qi et al., 2020). To control for the impact of the reacted
VOC concentration on SOA yields under different NOx conditions, Sarrafzadeh et al.
(2016) conducted experiments that maintained similar levels of VOC consumption.
However, due to differences in OH concentrations during photooxidation, there were
clear differences in the reaction times in each experiment. Under normal circumstances,
when exploring the impact of NOx on SOA yield, SOA yields are normalized based on
the reacted VOCs concentration, but this does not account for differences in the
consumption of VOCs. This approach inevitably overlooks the impact of semi-volatile
partitioning theory in SOA formation under different NOx conditions. The roles of
chemical processes are often considered due to the impacts of NOx on SOA yields, but
physical processes in SOA formation are equally significant and should be given more
attention.
α-Pinene is one of the most abundant monoterpene VOCs in the atmosphere
(Sindelarova et al., 2014; Guenther et al., 2012). Due to the high concentrations and
high SOA formation potential, α-pinene is one of the most important sources of SOA
in the atmosphere (Xu et al., 2015a; Zhang et al., 2018). In fact, α-pinene derived SOA
is commonly utilized in atmospheric models to represent biogenic SOA (Henry et al.,
2012). However, further in-depth research on the mechanisms of α-pinene SOA
formation is required to improve the fundamental basis and accuracy of atmospheric
models. In order to fully examine the effects of NOx on SOA formation, we used an



indoor chamber to investigate the yields of α-pinene derived SOA with different initial
NOx concentrations. The relationships among $NO_x$ concentration, VOCs consumption,
distribution of photooxidation products, and SOA mass yield were considered. Based
on the semi-volatile partitioning theory, this study aimed to better characterize the
mechanisms by which $NO_x$ affect SOA yield.

## 2. Experimental methods

### 2.1. Chamber studies

A series of α-pinene photooxidation experiments initiated by $NO_x$ were performed
in a temperature controlled photooxidation chamber. The chamber and its
characterization capabilities have been described in detail in our previous studies (Liu
et al., 2021a; Liu et al., 2021b). In brief, the photooxidation chamber was constructed
of Teflon-FEP film (0.06 mm) and surrounded with black light lamps (GE F40BLB) as
the light source for the photooxidation reaction. The black light lamps provided an
effective light intensity ($J_{NO2}$) of 0.165 $min^{-1}$ at full illumination. Before each
experiment, the chamber was cleaned by first evacuating all air and then filling it with
purified air. This filling-purging cycle was repeated 5 times between experiments to
ensure the residual particulate and α-pinene concentrations were less than 5 $cm^{-3}$ and
0.5 ppb, respectively. 5 $m^3$ zero air, which was supplied by a zero-air generator (111-
D3N, Thermo Scientific™, USA), was introduced into the chamber for the
photooxidation experiment. Analytically-pure, liquid α-pinene (Sigma-Aldrich) was
injected into a Teflon tube using a micro syringe, and then the α-pinene was evaporated
into gas and flushed into the chamber with zero air. NOx (Air Liquid Shanghai, 510 ppm
$NO_2$ in $N_2$) were introduced directly into the chamber. After all the reactants were well
mixed, photooxidation of α-pinene was initiated by turning the black light lamps on.
All the experiments were conducted in dry conditions with a relative humidity of 15 ±
5 %. Both OH and $O_3$ were formed under black light irradiation in the presence of NOx.
The detailed conditions for the α-pinene photooxidation experiments are listed in Table
1. The concentrations of α-pinene were kept as constant as possible across different





experiments to ensure the effects of NOx were not obscured. No seed particles were
used in the chamber experiments.

**Table 1.** Details of the α-pinene/NOx systems used in the photooxidation experiments.

| No. | α-pinene (ppb) | Δα-pinene (ppb) | NOx (ppb) | [VOCs]$_0$/[NOx]$_0$ | SOA mass conc.[a] ($\mu g\ m^{-3}$) | SOA yield |
|---|---|---|---|---|---|---|
| Exp. 1 | 116.2 | 67.3 | 12 | 9.7 | 26.0 | 6.5% |
| Exp. 2 | 117.2 | 96.2 | 25 | 4.7 | 43.8 | 7.7% |
| Exp. 3 | 117.8 | 115.3 | 68 | 1.7 | 54.3 | 8.0% |
| Exp. 4 | 119.3 | 117.6 | 150 | 0.8 | 34.6 | 4.9% |
| Exp. 5 | 116.9 | 114.9 | 337 | 0.3 | 15.3 | 2.2% |
| Exp. 6 | 115.6 | 112.8 | 600 | 0.2 | 11.4 | 1.7% |
| Exp. 7 | 258.3 | 70.7 | 8 | 32.3 | 20.9 | 5.0% |
| Exp. 8 | 251.0 | 168.7 | 26 | 9.7 | 144.0 | 14.5% |
| Exp. 9 | 263.4 | 234.0 | 52 | 4.1 | 370.3 | 26.2% |
| Exp. 10 | 250.8 | 245.3 | 113 | 2.2 | 347.6 | 23.7% |
| Exp. 11 | 247.8 | 244.2 | 237 | 1.5 | 253.5 | 17.1% |
| Exp. 12 | 260.0 | 256.0 | 369 | 0.7 | 226.4 | 14.6% |
| Exp. 14 | 254.2 | 249.9 | 669 | 0.4 | 175.0 | 11.8% |

[a] All mass concentrations were wall-loss corrected.

## 2.2. On-line instruments

The concentration of NOx was constantly monitored with a NO/NO$_2$/NOx analyzer
(Model 42C, Thermo Corporation, USA). The α-pinene concentration was measured
throughout the photooxidation process with a proton-transfer-reaction time-of-flight
mass spectrometer (PTR-tof-MS, Ionicon Analytik, Austria) using hydronium ($H_3O^+$)
ions. The drift tube of the PTR-tof-MS was operated at 60.0 °C ($T_{drift}$), 2.30 mbar ($P_{drift}$),
and 600V ($U_{drift}$). Three ions which are commonly observed in ambient air, i.e., $H_3O^+$
(21.0226 *m/z*), $NO^+$ (29.9980 *m/z*), and $C_3H_6O^+$ (59.0497 *m/z*), were used for PTR-tof-
MS mass calibration.
The formation and chemical characteristics of bulk SOA were measured with a
high-resolution time-of-flight aerosol mass spectrometer (HR-tof-AMS, Aerodyne
Research Inc., USA). During which, the AMS was only operated in the V-mode in order
to avoid W-mode data too noisy to analyze. The ionization efficiency (IE) of the AMS
was calibrated via monodisperse with dried ammonium nitrate (AN) aerosols (300 nm).





The SOA mass concentrations obtained from AMS measurements were compared and
corrected through a scanning mobility particle sizer. For SOA mass concentration
calculations, the relative ionization efficiencies (RIE) of 1.4 and 1.1 were applied for
organic compounds and nitrate, respectively (Middlebrook et al., 2012). The AMS data
were analyzed based on the standard applications of SQUIRREL 1.51H and PIKA
1.10H in Igor Pro (WaveMetrics, Inc., USA). $NO_2^+$($m/z$ = 46)/$NO^+$($m/z$ = 30) was used
to differentiate inorganic and organic nitrate in the AMS measurements (Xu et al.,
2015b; Kiendler-Scharr et al., 2016; Ng et al., 2017).

## 3.  Result and discussion

### 3.1 Effect of NOx on α-pinene SOA formation

The relationship between SOA yields and $NO_x$ is shown in Fig. 1. Here, SOA yield

was calculated as the SOA mass concentration divided by the reacted VOCs. Clearly,
the SOA yield increased rapidly at first, reached a maximum and then decreased
gradually with increasing initial $NO_x$ concentration under both high- and low-VOC
conditions. In this study, we defend the positive correlation between SOA yield and
NOx as low-NOx conditions, and a negative correlation of them as high-NOx
conditions.   In the low- and high-VOC experiments, the maximum SOA yields were
8.0% and 26.2% respectively. Since the change trend in SOA yield with $NO_x$ was not
affected by initial VOC concentration, the effect of $NO_x$ on SOA formation is discussed
in this section using the data from the low-VOCs experiments. The concentration of
VOCs in the chamber is generally higher than that in the real atmospheric environment.
Lower VOC concentrations would be closer to their levels in the actual atmosphere,
which would make experimental results more representative of the real atmospheric
environment.



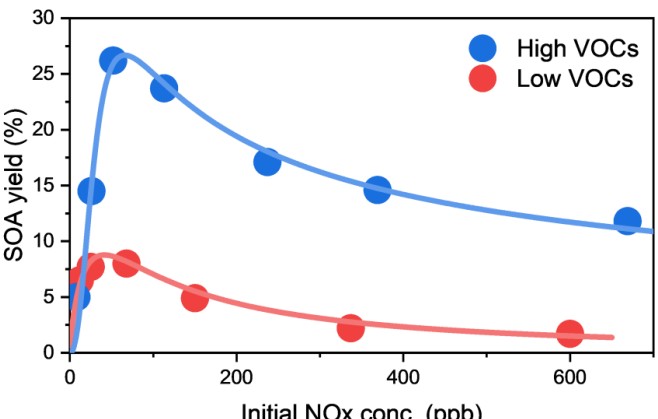

Figure 1. SOA yield from α-pinene photooxidation with different initial $NO_x$ concentrations under two levels of VOCs.

Like our study, similar relationships between SOA yields and initial $NO_x$ concentrations have been widely observed in previous studies (Aruffo et al., 2022; Liu et al., 2019b; Zhao et al., 2018; Lane et al., 2008; Ng et al., 2007; Kroll et al., 2006). $RO_2$ was the main intermediate in the photooxidation of VOCs and the concentration and fate of $RO_2$ depended on the $NO_x$ concentrations. $RO_2$ mainly reacted with NO and was rapidly converted to alkoxy radicals (RO) under high-$NO_x$ conditions. RO intermediates produce more high-volatility compounds than hydroperoxides (ROOH), which form through the reaction of $RO_2$ with $RO_2/HO_2$, and decrease SOA yields (Atkinson, 2000; Sarrafzadeh et al., 2016). Furthermore, the autooxidation of $RO_2$ can be inhibited through the $RO_2 + NO / NO_2$ reaction, and the reduction in HOMs further contributes to the decreased SOA yields under high-$NO_x$ conditions (Yu et al., 2022; Laskin et al., 2018).

However, the mechanisms by which SOA yields increase with increasing $NO_x$ concentrations under low-$NO_x$ conditions remain poorly understood (Camredon et al., 2007; Kroll et al., 2006). The atmospheric oxidizing capacity (AOC), which indicates the oxidizing ability of the atmosphere, is significantly influenced by NOx (Wang et al., 2023). In this study, we assessed AOC based on the decay ratio of VOCs ($AOC_t$ = -d



$[VOCs]_t$ / $[VOCs]_t$). The time-dependent consumption curves of VOC and the decay
ratios of VOCs under different NOx concentrations are shown in Fig. 2. All experiments
were allowed the same photooxidation time. For the low-NOx experiments, the
consumption of α-pinene increased with $NO_x$ concentration. Only when the initial $NO_x$
concentration was higher than 68 ppb was α-pinene completely consumed by the end
of the photooxidation period. The average decay ratio of VOCs increased from $4.75 \times 10^{-3}$
to $4.53 \times 10^{-2}$ as the initial NOx increased from 12 ppb to 150 ppb, and gradually
decreased to $2.64 \times 10^{-2}$ with an initial NOx of 600 ppb. Hence, the AOC in the chamber
also showed a trend of first increasing and then decreasing with increasing NOx
concentration. Sarrafzadeh et al. (2016) noted that the OH recycling reaction $NO + HO_2$
$\rightarrow NO_2 + OH$ was responsible for the $NO_x$-induced increase in OH concentration. Chen
et al. (2022b) revealed that the increasing NOx level obviously enhanced the
atmospheric oxidation ability in a NOx-sensitive (low-NOx) regime. The higher
consumption of VOCs and the faster VOC oxidation rate both suggested there was a
higher atmospheric oxidation ability in the reaction system (Ng et al., 2007). In addition,
NOx is an important sink for OH, so the competition between NOx and VOCs for OH
might have been responsible for the decreases in VOC consumption and AOC under
high-NOx conditions.

Although there were similar trends in the yield of SOA and AOC as the NOx

concentration increased, no correlation between SOA yield and AOC was observed in
our study (Fig. S1). It has been shown that increases in AOC are essential drivers of
increases in SOA mass concentration in the troposphere (Feng et al., 2019; Li et al.,
2023). The higher SOA mass concentration under high AOC conditions was due to
more VOCs being oxidized, rather than an increase in SOA yield. Therefore, the
increased AOC was not the direct mechanism by which increasing NOx concentration
influenced SOA yield.




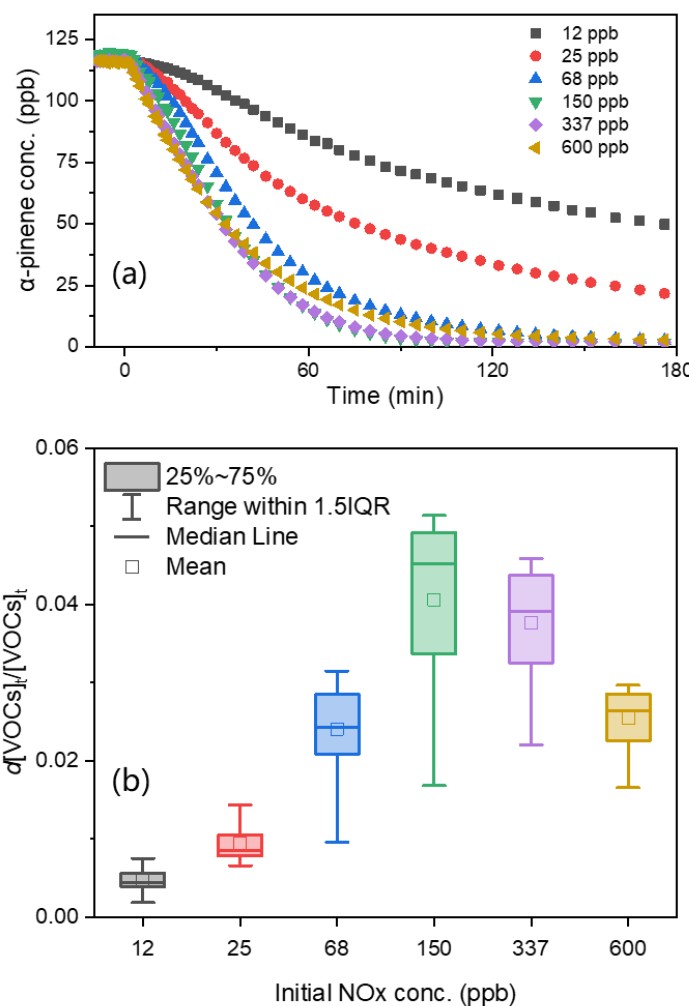

Figure 2. Changes in α-pinene concentrations over time (a) and the decay

ratios of the VOCs (b) with different initial $NO_x$ concentrations

The function of SOA yield with SOA mass concentration ($M_0$) was recalculated

based on Odum's SOA yield model (Fig. 3). In each experiment, an increase in $M_0$

directly resulted in an increase in SOA yield. Notably, if the photooxidation products

can be classified as low-volatility oxidation products, the SOA yield should remain

constant with increasing $M_0$ (Krechmer et al., 2015; Ehn et al., 2014; Odum et al., 1996).

However, the gas-particle distribution coefficients of semi-volatile substances are





directly related to their concentrations, and the distribution coefficients of semi-volatile
substances into the aerosol phase are larger at higher concentrations (Akherati et al.,
2019; Odum et al., 1996). Hence, the increasing distribution ratios of semi-volatile
organic products between aerosol and gas phases at high $M_0$ were responsible for the
increasing SOA yields with increasing photooxidation time (Kolesar et al., 2015;
Valorso et al., 2011; Takeuchi et al., 2022).
A two-product model is an effective method for fitting the relationship between
SOA yield and $M_0$ (Liu et al., 2019a; Odum et al., 1996). The model was calculated via
equation (1), shown below:

$$Y=M_0 \times \left( \frac{\alpha_1 K_{om,1}}{1+K_{om,1} M_0} + \frac{\alpha_2 K_{om,2}}{1+K_{om,2} M_0} \right) \tag{1}$$

where, $M_0$ is SOA mass concentration (mg m$^{-3}$); $\alpha_1$ and $\alpha_2$ are the mass-based
stoichiometric coefficients of species with low-volatility and semi-volatility products,
respectively; and $K_{om,1}$ and $K_{om,2}$ (m$^3$ μg$^{-1}$) are the gas-particle partitioning equilibrium
constants for low-volatility and semi-volatility products, respectively. The $\alpha_1$, $\alpha_2$, $K_{om,1}$,
and $K_{om,2}$ of α-pinene SOA formed under different initial NOx concentrations are shown
in Table 2, and the simulated SOA yields with changing $M_0$ are shown in Fig. 3.

Table 2 Parameters of the two product model for α-derived SOA under
different initial NOx concentration.

| Initial NOx conc. (ppb) | $\alpha_1$ | $K_{om,1}$ (m$^3$ μg$^{-1}$) | $\alpha_2$ | $K_{om,2}$ (m$^3$ μg$^{-1}$) |
|---|---|---|---|---|
| 12 | 0.048 | 0.19 | 0.28 | 0.0040 |
| 25 | 0.038 | 0.19 | 0.30 | 0.0039 |
| 68 | 0.028 | 0.19 | 0.32 | 0.0037 |
| 150 | 0.019 | 0.19 | 0.33 | 0.0031 |
| 337 | 0.017 | 0.19 | 0.35 | 0.0019 |
| 600 | 0.014 | 0.19 | 0.38 | 0.0016 |


It was clear that the curves of SOA yield to $M_0$ moved lower on the SOA yield axis
in Fig. 3 with increasing NO$_x$ concentration. According to Table 2, $\alpha_1$ decreased while
$\alpha_2$ increased as the initial NOx concentration increased. For the experiments conducted
with 12 ppb NOx, the $\alpha_1/\alpha_2$ ratio of α-pinene derived SOA was 0.17. This $\alpha_1/\alpha_2$ ratio
decreased to 0.13, 0.086, 0.058, 0.049, and 0.037 as the initial NOx concentration





increased to 25, 68, 150, 337, and 600 ppb, respectively. Lower $\alpha_1$ values indicate lower
proportions of low-volatility products in the SOA, while higher $\alpha_2$ values indicate
higher proportions of semi-volatility products in SOA. In addition, the $K_{om,2}$ value also
decreased, dropping continuously from 0.0040 to 0.0016 $m^3$ $\mu g^{-1}$ as the initial NOx
concentration increased from 12 to 600 ppb. This meant that the volatility of semi-
volatility products produced through $\alpha$-pinene photooxidation increased with initial
NOx concentration. Overall, with increasing initial NOx concentration, the low-
volatility products made up decreasing proportions (lower $\alpha_1$) of the total products
while semi-volatility products made up increasing proportions (higher $\alpha_2$) and higher
volatilities (lower $K_{om,2}$).

To determine why the volatility of SOA increase with increasing NOx, we

analyzed the consumption of VOCs under different NOx conditions (Fig. 2). Due to the
lower consumption rate of VOCs and low AOC, $\alpha$-pinene was not completely
consumed at the end of the photooxidation period under low-NOx conditions. Increased
consumption of VOCs will lead to higher concentrations of photooxidation products
generated in the chamber, so when the initial NOx concentration increased from 12 ppb
to 25 ppb to 68 ppb, the SOA mass concentration increased from 26.0 $\mu g$ $m^{-3}$ to 43.8
$\mu g$ $m^{-3}$ to 54.3 $\mu g$ $m^{-3}$. As mentioned above, SOA yield increased with the mass
concentration of SOA. Even if the fitting curve of the two-product model gradually
moved lower on the graph with increases in NOx, the SOA yield still increased from
6.5% to 8.0% when the initial NOx concentration increased from 12 ppb to 68 ppb.
Hence, under low-NOx conditions, more SOA was generated due to the increased AOC
and VOC consumption with increasing NOx. The enhancement of the SOA yield with
increasing NOx concentrations can be attributed to the increased ratio of the aerosol/gas
phase distribution resulting from higher concentrations of semi-volatile photooxidation
products. Chen et al. (2022c) compared the relative content of intermediate products
with different volatilities and found that the proportion of semi-volatile oxidized
products in gas-phase intermediate products was lower when experiments had higher
VOC consumption and SOA yields. Assuming that the proportions of different volatile



oxidation products remain constant, the smaller proportion of semi-volatile organic
products in the gas phase suggested that a larger proportion of semi-volatile organic
products was condensed into the particulate phase when more VOCs were consumed.
This result also supported our conclusion that the increased aerosol/gas phase
distribution ratio of semi-volatile products was the dominant driver underlying the
enhanced SOA yields with increased VOC consumption under low-NOx conditions.

It should be noted that if the SOA yield is simulated based only on the two-product

model, the SOA yield should increase by 34% and 51% when NOx increases from 12
ppb to 25 ppb and 68 ppb, respectively. However, in reality, the SOA yield only
increased 18% and 23%, respectively (Fig. S2). The real SOA yields were not only
lower than the simulated yields, but the discrepancy between the real and simulated
SOA yields increased with increasing NOx concentration. As mentioned earlier,
reactions between $RO_2$ and NOx form highly volatile oxidized compounds. Therefore,
the increasing discrepancy between real and simulated SOA yields with increasing NOx
concentration indicated that inhibition of SOA formation through the $RO_2$ + NOx
reaction pathway was always occurring. This indicated that, under low-NOx conditions,
the increase in SOA yield resulted from a combination of effects, the positive effect of
the increased aerosol/gas phase distribution ratios of semi-volatile products and the
negative effect of the $RO_2$ + NOx reaction.



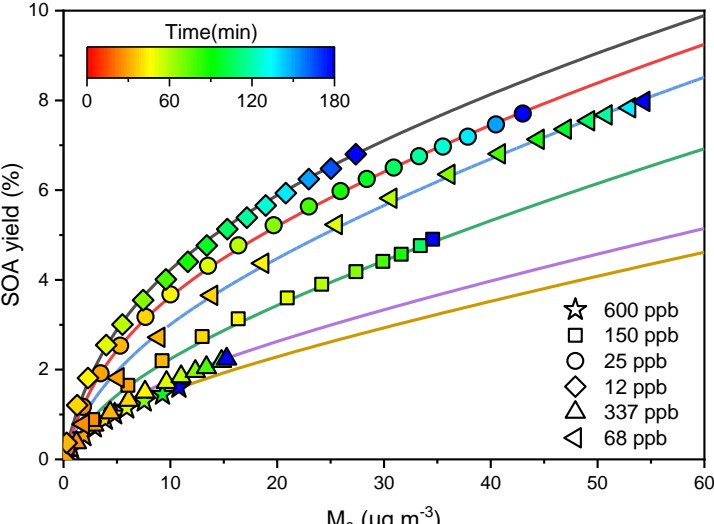

Figure 3. SOA yields as a function of organic aerosol mass concentration $M_0$
of α-pinene at different initial $NO_x$ concentrations. The simulated SOA yields
based on the two-product model are shown by the solid lines.

The effects of NOx on the yields and chemical properties of SOA have been well documented in previous studies (Chen et al., 2020; Eddingsaas et al., 2012; Sarrafzadeh et al., 2016; Zhao et al., 2018). However, few studies have focused on how $NO_x$ influences VOC consumption during photooxidation, so the relationship between the concentrations of $NO_x$ and oxidation products has not been fully elucidated. For SOA, which is mainly composed of semi-volatile oxidation products, the gas-particle distribution coefficients of semi-volatile substances are directly related to their concentrations. Therefore, SOA yield can be affected by the concentrations of semi-volatile oxidation products, and this should be taken into consideration when studying SOA formation process. Hence, future studies of photooxidation and SOA formation should pay more attention to the formation processes of semi-volatile substances.

## 3.2 SOA chemical composition at different $NO_x$ concentrations

SOA chemistry must be studied at the molecular level to better understand the characteristics of SOA formation. The bulk chemical properties of SOA generated





under low- and high-NO$_x$ concentrations are shown in Fig. 4.

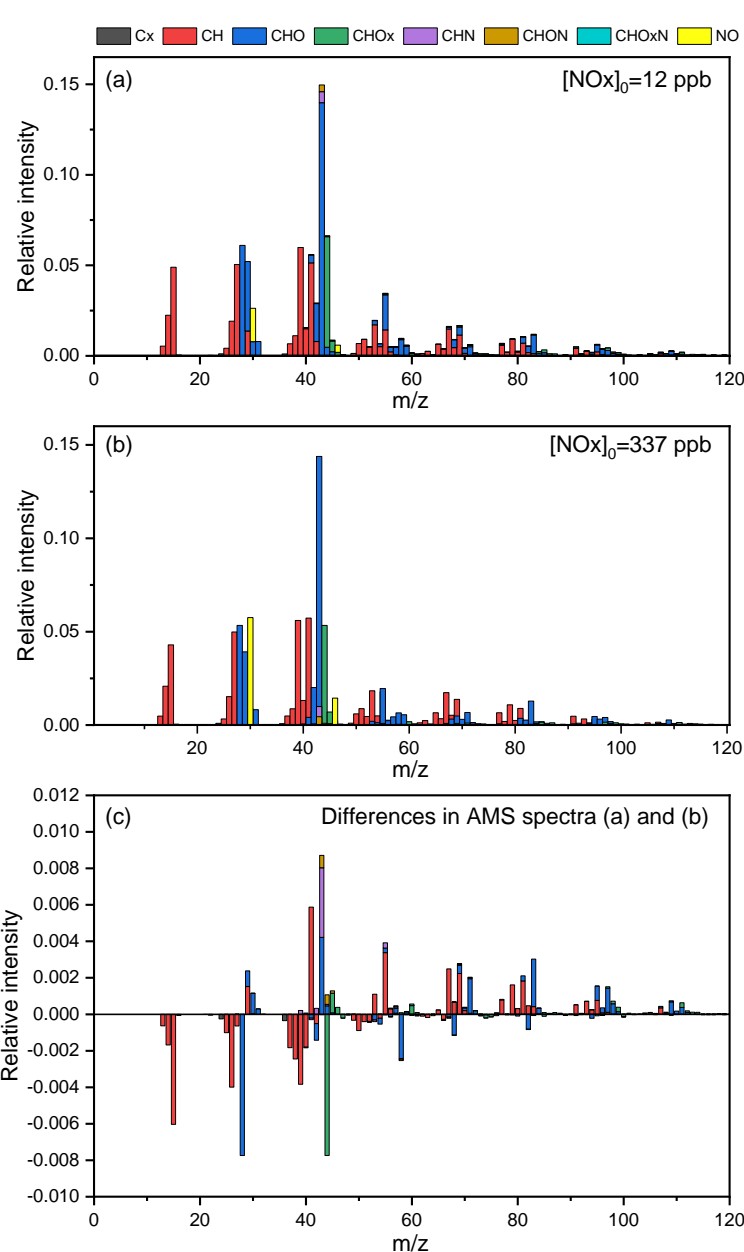


Figure 4. Typical AMS spectra of α-pinene SOA formed under low-NO$_x$
conditions (a), high-NO$_x$ conditions (b), and the differential spectra comparing
the low- and high-NO$_x$ conditions (c).



Organic fragments of $C_2H_3O^+$ ($m/z$ = 43), which originated from the fragmentation
of aldehydes and ketones, were the dominant peaks in the AMS. The strong organic
AMS signal at $m/z$ = 43 ($f_{43}$) indicated that the SOA oxidation level was relatively low.
The intensity of the $CO_2^+$ ($m/z$ = 44), representing the thermal decarboxylation of
organic acid groups, was rather low, which indicated there was low carboxylic acid
content in the α-pinene derived SOA. Carboxylic acid has a relatively low volatility and
enters the particulate phase relatively easily, which means that the carboxylic acid
produced through the photooxidation of α-pinene was likely less than that detected in
the AMS. In our study, the $f_{43}$ and $f_{44}$ of α-pinene SOA ranged from 0.160 to 0.175 and
from 0.069 to 0.074, respectively. According to the "triangle plot" of the AMS, the SOA
derived from α-pinene photooxidation mainly fell in the lower area designated semi-
volatile oxygenated organic aerosols (SV-OOA) (Ng et al., 2010). The AMS results
provided direct evidence that semi-volatile products were the main components of α-
pinene SOA formed through $NO_x$ photooxidation. This result further supported our
observations in Section 3.1, wherein the aerosol/gas-phase distribution ratio of
photooxidation products increased with increasing SOA mass concentration and VOC
consumption, resulting in the enhancement of SOA yield with increasing $NO_x$ at low-
NOx conditions.
The differences in the AMS spectra between low- and high-NOx conditions are
compared in Fig. 4 (c). Although the inhibitory effect of the reaction between NOx and
$RO_2$ on SOA formation was confirmed, the chemical compositions of the SOA formed
under different NOx conditions showed only minor differences. Photooxidation
products formed through the reaction of NO with $RO_2$ radicals usually have relatively
high volatilities. However, the α-pinene derived SOA was mainly composed of semi-
volatile photooxidation products, and these photooxidation products formed under
high-NOx conditions with higher volatilities may not easily be condensed into the
particulate phase, so they were mainly present in the gas phase. This means that the
high volatility products formed via the $RO_2$ + NO reaction path under high-NOx
conditions did not affect the chemical composition of SOA. Therefore, the AMS spectra





of the α-pinene SOA were not significantly different between low- and high-NOx
conditions.

In addition to the generation of RO, the reaction of $RO_2$ with NO can also form N-

containing compounds (NOCs). The NOC contents in SOA under different NOx
conditions were calculated based on the AMS fragments of $NO^+$ and $NO_2^+$. $NO^+$ and
$NO_2^+$ are characteristic fragments of NOCs, but the same $NO^+$ and $NO_2^+$ fragments can
also be detected by AMS in the forms of inorganic nitrates. In order to distinguish
organic nitrates from total nitrates, we used the same method as described in our
previous study based on the differences in the ratios of $NO^+/NO_2^+$ for organic and
inorganic nitrates in the AMS mass spectra (Xu et al., 2015b; Ng et al., 2017; Day et
al., 2022). The concentrations of $NO^+$ and $NO_2^+$ from organic nitrates were calculated
using the following equations:

$$NO_{2,org} = \frac{NO_{2,meas} \times (R_{meas} - R_{AN})}{R_{ON} - R_{AN}} \tag{4}$$

$$NO_{org} = R_{ON} \times NO_{2,org} \tag{5}$$

where $R_{meas}$ is the $NO^+/NO_2^+$ ratio from the AMS results; $R_{ON}$ and $R_{AN}$ are the
$NO^+/NO_2^+$ ratios for organic nitrate and ammonium nitrate, respectively; and $NO_{2,meas}$
represents the total $NO_2^+$ fragments obtained from the AMS data. In this study, $R_{AN}$ was
about 1.1 based on the measurements and $R_{ON}$ was assumed to be about 10 referring to
the study by Takeuchi and Ng (2019). Previous studies have shown monoterpene
hydroxyl nitrate (m/z = 215), pinene keto nitrate (m/z = 229), and monoterpene
dicarbonyl nitrate (m/z = 247) to be the main NOCs from α-pinene photooxidation
(Chen et al., 2022a; Li et al., 2018). Here, the formation of NOCs was calculated based
on organic nitrate contents and the molecular weights of NOCs.

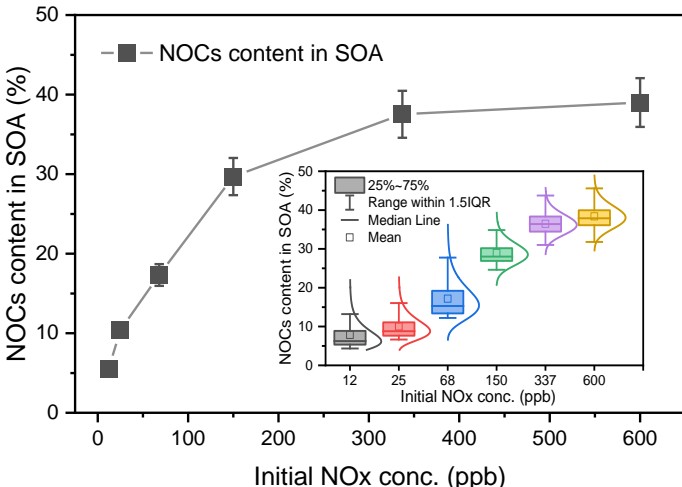

Figure 5 Content and mass concentration of NOCs under different initial NOx conditions.

The contents and mass concentrations of NOCs under different NOx levels are

shown in Fig. 5. The increase in NOC contents with NOx could be divided into two
stages. The first stage occurred when the initial NOx concentration increased from 12
ppb to 150 ppb. During this stage the content of NOCs in the SOA increased linearly,
indicating that the NOx was the limiting factor for NOC formation in low-NOx
conditions. The second stage occurred as NOx concentrations continued to rise to 600
ppb. During this stage, the growth in NOC content gradually slowed while approaching
the maximum value. Based on the nonlinear fit between NOx concentration and NOC
content, the maximum value of NOCs content in SOA was predicted to be about 39 $\pm$
3.8%. This indicated that, under extremely high-NOx conditions, NOCs could account
for up to two-fifths of the α-pinene SOA. These results not only showed that NOCs play
a more important role in biogenic SOA formation under high-NOx conditions, but can
also serve as a basis for estimating the generation of NOCs in SOA.
**3.3 Effect of initial VOC concentration on SOA formation**

The SOA yields with different α-pinene concentrations were also compared in this

study. As shown in Fig. 1, the change trends in SOA yields with increasing NOx



concentrations were similar across all experiments, including low- and high-VOC
conditions. However, higher initial VOC concentrations led to increases in both
maximum SOA yield and the initial NOx concentration required for the SOA yield to
reach its maximum value. Under low-VOCs conditions, based on the no-linear fitting
of SOA yield with initial NOx, the peak SOA yield was 8.8% when the initial $NO_x$
concentration was 41 ppb. The maximum SOA yield increased to 26.7% under high-
VOC conditions, and the concentration of NOx corresponding to the maximum SOA
yield increased to 66 ppb. Under high-VOC conditions, the maximum SOA mass
concentration was 370.3 μg m$^{-3}$. However, the maximum mass concentration of SOA
formed under high-VOC conditions was only 54.3 μg m$^{-3}$, which was 6.8 times lower
than that under high-VOCs.
Previous studies have suggested that the formation of SOA from α-pinene
photooxidation is primarily limited by NOx concentration under low-NOx conditions,
and variation in VOC concentrations have little effect. However, as shown in Fig. 1, the
SOA yield under high-VOC conditions was consistently higher than that under low-
VOC conditions. Moreover, the ratio and gap in SOA yields between high- and low-
VOC conditions significantly increased with increasing NOx concentrations when NOx
concentrations were lower than 60 ppb (Fig. S3). Therefore, it can be concluded that
under low-NOx conditions, the formation of SOA from α-pinene photooxidation is
limited by both VOCs and NOx concentration. Based on differences in VOC
consumption between high- and low-VOCs conditions (Fig. S4), it appeared that, at the
same NOx concentration, the consumption of VOCs was higher under high-VOC
conditions and the difference in VOC consumption between high- and low-VOC
conditions gradually increased with increasing NOx. According to the aerosol/gas-
phase distribution ratio of semi-volatile products described in Section 3.1, higher VOC
consumption can generate more semi-volatile oxidation products, which can enhance
the SOA yield. The continuously increasing concentration gradient in VOC
consumption between high- and low-VOC conditions indicated that the SOA yield
under low-NOx conditions was also affected by the concentration of VOCs.



When the initial NOx concentration was higher than 100 ppb, the growth rate of
the SOA yield ratio between high- and low-VOC conditions was 67.7% lower than that
under low-NOx conditions. The ratio of SOA yield from high-VOC experiments was
about 3–8 times higher than that from the low-VOC experiments, which surpassed the
VOC ratio between different VOC conditions. This indicated that, under high-NOx
conditions, SOA yield has a high sensitivity to changes in VOC concentrations, and the
inhibition effect of reducing VOCs on SOA yields would be more pronounced under
high-NOx conditions. It should be noted that, while the difference in VOC consumption
did not change further under high-NOx conditions, the ratio of the SOA yields between
high- and low-VOCs conditions continued to increase with increasing NOx
concentrations. At lower VOC conditions, the $RO_2$ + NO pathway was more
competitive, which enhanced the production of high-volatile products compared to in
the higher VOC experiments. This sustained increase in the ratio of SOA yields may
have been mainly due to the differences in volatilities of the oxidation products
produced under high- and low-VOC conditions. Therefore, as illustrated in Fig. S3, it
can be inferred that changes in VOC concentrations have a more pronounced effect on
SOA yield under higher NOx conditions.
Although the trend in SOA yields with increasing NOx was similar under different
VOC conditions, the change trend in AOC with increasing NOx was different (Fig. S5).
First, under low-NOx conditions, the AOC under high-VOC conditions was lower than
when under low-VOCs. The consumption of α-pinene was higher in the high-VOC
experiments than in the low-VOC experiments and more oxidants were consumed by
α-pinene under high-VOC conditions, which may have led to the inhibition of AOC.
Secondly, AOC only increased with increasing NOx concentrations under high-VOC
conditions. According to the fitting results in Fig. S5, under low-VOC conditions, AOC
initially increased with NOx and then decreased, with an inflection point at VOCs/NOx
≈ 0.47. However, under high-VOC conditions, AOC increased with initial NOx
concentration across the entire NOx gradient. Although the rate of change in AOC
gradually slowed as NOx increased, no inflection point occurred in the AOC change



trend even when NOx reached 669 ppb, at which point VOCs/NOx decreased to 0.38.
These results suggested that AOC is jointly affected by NOx and VOCs, and that
VOCs/NOx is not sufficient as a direct basis for the evaluation of AOC. This
demonstrated the complicated nature of accurately simulating AOC, which is an
important research topic that needs to be studied further.

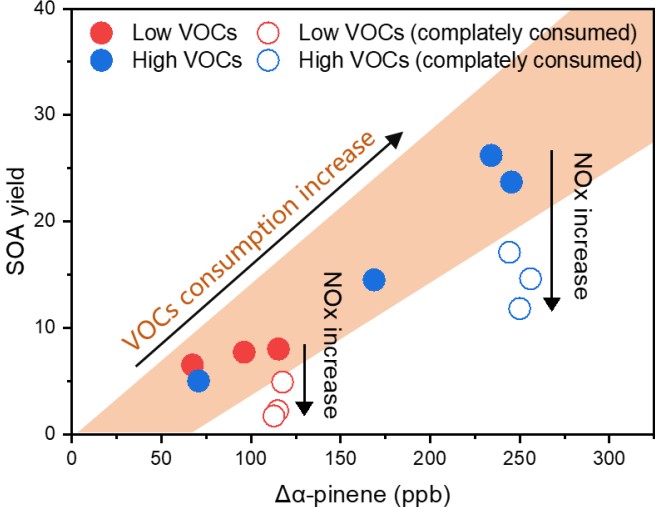

Figure 6 Relationship between consumed VOCs and SOA yields under different NOx
and VOC conditions

The SOA yield trends at different AOCs are shown in Fig. S1. Based on the
photooxidation experiments with different VOC concentrations, there was no clear
correlation between SOA yield and AOC. This result further explains why increases in
AOC are not the direct cause of increased SOA yields from α-pinene photooxidation.
We further compared the consumption of VOCs with SOA yield (Fig. 6) and found a
significant positive correlation between the two. This result was consistent with the
study by Jiang et al. (2022) which reported that SOA yield increased with increasing
consumption of VOCs, even when the OH concentration was held constant across
experimental photooxidation groups. The continuous growth trend in SOA yield with
increasing consumption of VOCs, which overlapped among different VOC conditions,
further supported the processes discussed in Section 3.1; that is to say, changes in the
aerosol/gas-phase distribution ratio for semi-volatile products with changing VOC



concentrations lead to increased SOA yields with increasing NOx concentration under
low-NOx conditions.
However, when the NOx concentrations exceeded 68 ppb and 113 ppb under low-
and high-VOC conditions, respectively, α-pinene was almost entirely consumed during
the photooxidation process. This meant that, although AOC continued to increase with
increasing NOx, further increases in AOC only accelerated VOC consumption time
without impacting total VOC consumption or photooxidation products formation. As a
result, the SOA yield did not continue to increase. In contrast, more NOx would
promote the $RO_2 + NO$ reaction, which would result in a decreasing trend in the SOA
yield.

## 486 4. Atmospheric implications

$NO_x$ is a key substrate for photooxidation in the atmosphere. Changes in $NO_x$
concentrations can lead to significant changes in the photooxidation of VOCs and affect
subsequent SOA formation. In this study, SOA yields were observed to increase sharply
at first, and then decrease gradually with increasing $NO_x$ concentration. The maximum
SOA yields in the low- and high-VOC conditions were 8.0% and 26.2%, respectively.
Based on the relationship between SOA yield and $M_0$, as well as the SOA chemical
composition, semi-volatile oxidation products were the main components of α-pinene
derived SOA. Under low-$NO_x$ conditions, increasing $NO_x$ concentration increased the
consumption of VOCs during photooxidation and lead to an increase in the SOA yield.
We concluded that the combination of increased distribution ratios of the semi-volatile
oxidation products with increasing amounts formed was key in the promotion of SOA
yield with increasing NOx. Conversely, the effect of NOx on AOC did not directly cause
the increase in SOA yield. NOx can significantly increase the content of NOCs in SOA.
At their highest, under extremely high-NOx concentrations, NOC contents made up as
much as two-fifths of α-pinene derived SOA.
With the progression of urbanization and the changes to the natural environments
surrounding urban areas, atmospheric environments of have become quite different (Xu



et al., 2015b; Domínguez-López et al., 2014; Shon et al., 2007; Agbo et al., 2022). Most
rural areas exhibit $NO_x$ limitation, while urban areas more often exhibit VOC limitation
(Tan et al., 2018; Santos et al., 2021; Hui et al., 2018). In areas with low-NOx
concentrations, the consumption of biogenic VOCs might not be significant despite
there being high biogenic VOC emissions, which would limit the formation of SOA.
However, if unreacted VOCs are transported to areas with high-$NO_x$ concentrations (i.e.,
from rural to urban areas), or vice versa, there could be explosive increases in SOA
concentrations and noticeable changes of SOA chemical compositions. Therefore, the
regional transport of pollutants between urban and forested areas will significantly
impact the formation of atmospheric aerosols and environmental pollution. This was
supported by our study, which demonstrated how human activities can enhance SOA
formation from biogenic emissions.
Finally, as a typical goal, research often works toward creating or improving
models to simulate atmospheric phenomena. However, past simulations have generally
underestimated SOA concentrations when compared to field observations, especially
during pollution periods (Ling et al., 2022; Kelly et al., 2018). Our study noted that,
under high-NOx and high-VOC conditions, SOA yield exhibited significant changes
with changes in reaction conditions (Fig. S3). Under polluted conditions, the faster
consumption of VOCs might lead to underestimates of VOC concentrations in the
atmosphere and, subsequently, the generation of oxidation products, ultimately
resulting in underestimates of SOA yields. Hence, our research provides crucial
information for improving the accuracy of air quality models simulating SOA formation.

**Data availability**
The data used to support the conclusions in this study are available at a public data
repository of Figshare via https://figshare.com/articles/dataset/NOx_photooxidation_-
pinene/25200929 (S. Liu, 2024)

**Author contributions**



SL and GW designed the whole work and wrote the paper. SL did the experiment, collected the samples, conducted the sample analysis and performed the data interpretation. All authors contributed to the paper with useful scientific discussions and comments.

## Competing interests

The authors declared that they have no conflict of interest.

## Acknowledgements

This work was funded by the National Natural Science Foundation of China (No. 42130704, U23A2030), the China Postdoctoral Science Foundation (No. 2022T150215), and ECNU Happiness Flower Project.

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
