# Peer review of "Changes in Aerosol/Gas-Phase Distribution Ratio of Semi-Volatile Products Affect Secondary Organic Aerosol Formation with NOx from α-Pinene Photooxidation"

_EGUsphere, 2024_

## Referee Comment (RC1)

**Comments for egusphere-2024-1599**

This paper studies the formation of secondary organic aerosol from photo-oxidation of α-pinene reacting with NOx. Experiments were conducted in batch mode in a simulation chamber. The SOA yields were investigated under varying concentrations of α-pinene and NOx. The authors discussed the SOA yields as a function of organic aerosol mass concentration from the site of aerosol/gas-phase distribution ratio of semi-volatile products. In addition, SOA chemical composition at the bulk level for different NOx concentrations was also studied.

There is merit to the topic of SOA formation from monoterpene oxidation in the presence of NOx. However, several major and minor comments need to be addressed before the manuscript can be considered for publication.

**General comments:**

1. As the authors mentioned in the manuscript, this study uses unrealistically high VOC and NOx concentrations which are much higher than in the general atmosphere. I am therefore wondering how the resulted chemical regime of the experiments could have affected the results presented and their implication to the real atmosphere. Can authors discuss more about the atmospheric implication of the results?

2. Also, there have been a large amount of investigations on the SOA formation from α-pinene oxidation during the past decades. What is the novelty of this study?

3. The effect of 'Aerosol/gas-phase distribution ratio of semi-volatile products' is one of the major points in this work. However, there is a lack of clear identification and quantification of 'Aerosol/gas-phase distribution ratio of semi-volatile products' in the manuscript. Can the authors explain it in more detail in the manuscript/supplement? Besides, the identification of 'semi-volatile products' is also not clear. Do they refer to the compounds in any range of saturation vapor pressures, or any range of partitioning coefficients, or other specific identification in this work? Or did I miss anything?

4. The whole section of '3.2 SOA chemical composition at different NOx concentrations' is based on the calculation of N-containing compounds (NOCs). The authors use a constant $R_{ON}$ value of 10 to quantify the NOCs by the equations 4 and 5. This value is from Takeuchi and Ng (2019). However, the VOC and NOx concentrations in this work are largely different from the concentrations in the study of Takeuchi and Ng (2019). This may lead to different chemical composition of generated SOA between two works, which may affect the $R_{ON}$ value (Xu et al., 2015). Also, the chemical compositions of SOA in different experimental conditions are mentioned to be different in this work. Thus, using a constant $R_{ON}$ value for all experiments may lead to uncertainties in the quantification of NOCs. Can the authors discuss how the resulted chemical composition could have affected the quantification of NOCs (especially the major results showed in Figure 5)?

5. I would suggest authors to proofread the manuscript and check the grammar in English. Especially, terminology should be correctly used. Self-identified terms will need to be fully explained at the first time.

**Specific comments:**
1. Line161. The calculation of yields belongs to the section of Experimental Methods.
2. Line 200. What is the unit of decay ratio?
3. Are the results corrected by the effect of dilution?
4. The quantification of the particle concentration is important in this work. Can authors show the simultaneous measurements of SMPS, and state also the collection efficiency used for AMS measurement? What is the uncertainty of the yields in Table 1?
5. Line230. Does 'gas-particle distribution coefficients' mean gas-particle partition coefficients?
6. The period in each experiment for yields determination is not clear. This could be important when comparing the different SOA in batch-mode studies.
7. Line232. 'into the aerosol phase are larger at higher concentrations' of particle phase? The statement is not clear.
8. The calculation of the two-product model (line237-248, equation 1, Tabe 2) should be moved into the section of Experimental Methods.
9. Line259-261. The sentence is not clear. The results do not show any change of the volatility. How can it be concluded that 'the volatility of semi volatility products produced through α-pinene photooxidation increased with initial NOx concentration'?
10. Line266-275 repeat that yields increase with increasing NOx concentration. Please get them streamlined and delete the unnecessary repeats. Further, why the increasing yields with increasing NOx are interpreted to be the contribution of higher contributions of semi-volatile products? Do authors mean higher NOx concentrations lead to a higher fraction of semi-volatile products among total products than at lower NOx concentrations?
11. Line283-286. What do the 'volatile oxidation products' refer to? Oxidation products in semi volatility or in all volatility groups? Do the proportions refer to the amount of total semi-volatile oxidation product in the sum of all gas- and particle-phase oxidation products? These confuse me. Also, I would suggest authors to give citations and explain why the assumption given here is correct.
12. Line367-373. The calculation belongs to the section of Experimental Methods.

**Technical comments:**
1. line60. Abbreviations should be identified when they show up for the first time.
2. Figure6. In the legend, 'complately' is incorrect.

**References**

Takeuchi, M., and Ng, N. L.: Chemical composition and hydrolysis of organic nitrate aerosol formed from hydroxyl and nitrate radical oxidation of α-pinene and β-pinene, Atmos. Chem. Phys., 19, 12749-12766, 10.5194/acp-19-12749-2019, 2019.

Xu, L., Suresh, S., Guo, H., Weber, R. J., and Ng, N. L.: Aerosol characterization over the southeastern United States using high-resolution aerosol mass spectrometry: spatial and seasonal variation of aerosol composition and sources with a focus on organic nitrates, Atmos. Chem. Phys., 15, 7307-7336, 10.5194/acp-15-7307-2015, 2015.

---

## Author Comment (AC1)

**Reviewer #2**

This paper studies the formation of secondary organic aerosol from photo-oxidation of α-pinene reacting with NOx. Experiments were conducted in batch mode in a simulation chamber. The SOA yields were investigated under varying concentrations of α-pinene and NOx. The authors discussed the SOA yields as a function of organic aerosol mass concentration from the site of aerosol/gas-phase distribution ratio of semi-volatile products. In addition, SOA chemical composition at the bulk level for different NOx concentrations was also studied.

**There is merit to the topic of SOA formation from monoterpene oxidation in the presence of NOx.** However, several major and minor comments need to be addressed before the manuscript can be considered for publication.

**General comments:**

1.

As the authors mentioned in the manuscript, this study uses unrealistically high VOC and NOx concentrations which are much higher than in the general atmosphere. I am therefore wondering how the resulted chemical regime of the experiments could have affected the results presented and their implication to the real atmosphere. Can authors discuss more about the atmospheric implication of the results?

**Author reply:**

We are very grateful to the reviewer for this comment.

We add the sentences of "Though the initial concentrations of VOCs were higher than that in the atmosphere and the SOA mass loadings was vastly overrated, this study could provide new insights for the nonlinear relationship of NOx with SOA yield, and may be informative to future studies with more atmospheric-relevant concentrations of reactants. And then, according to the changes in the aerosol/gas-phase partition ratio for semi-volatile products with changing VOC concentrations, the ratio of semi-volatile oxidation products distributed in the gas phase would be much higher, but the ratio of that in particulate phase is lower in the real atmosphere than that in our study. The identification of semi-volatile oxidation products in both aerosol and gas phase would further promote the understanding on the process of SOA formation. In addition, the

higher ratio of semi-volatile oxidation products distributed in the gas phase in the atmosphere leads to a more pronounced influence of environmental factors on SOA formation, such as acid-catalyzed heterogeneous reactions, liquid-phase reactions, compared to the laboratory studies." in the end of Section 3.3.

2.

Also, there have been a large amount of investigations on the SOA formation from α-pinene oxidation during the past decades. What is the novelty of this study?

**Author reply:**

We are very grateful to the reviewer for this comment.

In previous studies, the SOA yield, which used for the analyzation, is obtained in the end of the photooxidation. Odum model simulations in the previous studies were based on multiple sets of experiments. The simulation of SOA formation process in a single set of experiments at different times were relatively rare. Therefore, the effect of aerosol/gas-phase distribution in the SOA formation processes is often overlooked. In this study, we analyzed the dynamic changes of SOA formation based on a variety of online instruments.

Additionally, although the SOA yield increases initially and then decreases as the NOx concentration increases is often mentioned, but the reason for the increase of SOA yield in low NOx conditions have not been well explained. Sarrafzadeh et al. (2016) and Qi et al. (2020) pointed out that the promotion of $NO_2$ on SOA yield was due to the increase of OH concentration in the chamber (Sarrafzadeh et al., 2016;Qi et al., 2020). Based on this explation, the increased oxidizing ability with initial NOx concentration would lead to a greater consumption of VOCs. The increased consumption of VOCs VOC emissions lead to an increase in the formation of SOA, but it cannot effectively explain why the SOA yield was increased accordingly. In addition, as far as we know, some studies on SOA yield have found that NOx has only an inhibitory effect on SOA yield (Jiang et al., 2019;Sarrafzadeh et al., 2016;Zhao et al., 2018). The different trends in SOA yield with NOx also suggest that the study on the influence of NOx on SOA formation is not complete.

In our study, the dynamic evolution of SOA yield in each photooxidation process single experiment, the aerosol/gas-phase distribution ratio of semi-volatile products can effectively explain the phenomenon of increasing SOA yield with rising NOx levels, and also identify the reasons for different trends of SOA yield with NOx concentration in the previous studies.

3.

The effect of 'Aerosol/gas-phase distribution ratio of semi-volatile products' is one of the major points in this work. However, there is a lack of clear identification and quantification of 'Aerosol/gas-phase distribution ratio of semi-volatile products' in the manuscript. Can the authors explain it in more detail in the manuscript/supplement? Besides, the identification of 'semi-volatile products' is also not clear. Do they refer to the compounds in any range of saturation vapor pressures, or any range of partitioning coefficients, or other specific identification in this work? Or did I miss anything?

**Author reply:**

We are very grateful to the reviewer for this comment.

We identified the volatile of SOA based on the "triangle plot" of $f_{43}$ and $f_{44}$ which obtained by AMS in Line 336-339: "In our study, the $f_{43}$ and $f_{44}$ of α-pinene SOA ranged from 0.160 to 0.175 and from 0.069 to 0.074, respectively. According to the "triangle plot" of the AMS, the SOA derived from α-pinene photooxidation mainly fell in the lower area designated semi-volatile oxygenated organic aerosols (SV-OOA) (Singh et al., 2019;Reyes-Villegas et al., 2016;Hao et al., 2014;Ng et al., 2010). The AMS results suggested that the α-pinene SOA formed through NOx photooxidation exhibited semi-volatile characteristics."

According to the Odum two-product model, the fitted curves for each experiment gradually moved to the lower position with the increase of NOx, which indicated that the volatility of photooxidation products increases with the increasing initial NOx concentration. At the same time, the higher NOx concentration enhances the RO reaction path, which also indicates the formation ability of SOA was inhibited by NOx. But in low NOx conditions, the consumption of VOCs and the SOA mass concentration was increased with initial NOx concentrations. Based on the nature of Odum model,

the SOA yield was increases with increasing SOA mass concentration in each photooxidation experiment through the gradual increase in the gas-particle partition coefficients of photooxidation product (Odum et al., 1996). We have described the change of gas-particle partition coefficients of photooxidation products for each experiment as "However, the gas-particle partition coefficients of semi-volatile substances are directly related to their concentrations, and the partition coefficients of semi-volatile substances into the particulate phase are larger at higher concentrations (Akherati et al., 2019;Odum et al., 1996). Hence, the increasing partition ratios of semi-volatile organic products between aerosol and gas phases at high $M_0$ were responsible for the increasing SOA yields with increasing photooxidation time (Takeuchi et al., 2022;Kolesar et al., 2015;Valorso et al., 2011)."

The inhibition of SOA formation by the roles of chemical processes (the branching of RO2 react with RO2/HO2 or NO) and facilitation of SOA formation by the physical processes (aerosol/gas-phase distribution) jointly effect the SOA yield. With the increase of NOx concentration, more SOA was formed. The SOA mass concentration was increased from 26.0 μg m$^{-3}$ to 54.3 μg m$^{-3}$ as the initial NOx concentration increased from 12 ppb to 68 ppb. Hence, we believe that the increased SOA yield with increasing NOx concentration is responsible by the aerosol/gas-phase partition of semi-volatile products.

In addition, the relative content of different classes volatile products in both gas and particulate phase observed in the studies of Chen et al. (2022c) also show an increasing aerosol/gas-phase partition of semi-volatile products with the increasing VOC consumption and SOA yield.

4.

The whole section of '3.2 SOA chemical composition at different NOx concentrations' is based on the calculation of N-containing compounds (NOCs). The authors use a constant RON value of 10 to quantify the NOCs by the equations 4 and 5. This value is from Takeuchi and Ng (2019). However, the VOC and NOx concentrations in this work are largely different from the concentrations in the study of Takeuchi and Ng (2019). This may lead to different chemical composition of generated SOA between two works, which may affect the RON value (Xu et al., 2015). Also, the chemical

compositions of SOA in different experimental conditions are mentioned to be different in this work. Thus, using a constant RON value for all experiments may lead to uncertainties in the quantification of NOCs. Can the authors discuss how the resulted chemical composition could have affected the quantification of NOCs (especially the major results showed in Figure 5)?

**Author reply:**

We are very grateful to the reviewer for this comment.

For the $NO_x^+$ ratio method, considering the large variation in $NO^+/NO_2^+$ ratio for different organic nitrates, the largest uncertainty is associated with the value of RON. this is challenging to determine the RON value for every organic nitrate species. Based on the previous studies, the RON values of 5 and 10 likely correspond to upper and lower bounds of the $NO_{3,org}$ concentrations estimated by the $NO_x^+$ ratio method.

The sentence of "and RON was assumed to be about 10 referring to the study by Takeuchi and Ng (2019)." in Line 372 is fixed as "Considering the large variation in $NO^+/NO_2^+$ ratio for different organic nitrates, the $R_{ON}$ values were assumed to be 5 and 10 as the upper and lower bounds referring to the previous studies (Takeuchi and Ng, 2019;Xu et al., 2015)." in the revised manuscript in Line 187-190.

The Figure 5 is changed as

[Figure]

The sentence of "During this stage, the growth in NOC content gradually slowed while approaching the maximum value. Based on the nonlinear fit between NOx concentration and NOC content, the maximum value of NOCs content in SOA was predicted to be about 39 ± 3.8%." in Line 388-391 is fixed as "During this stage, the growth in NOC content gradually slowed while approaching the maximum value. Based on the nonlinear fit between NOx concentration and NOC content, the maximum value of NOCs content in SOA was predicted to be in the range of 39% to 48%." in the revised manuscript in Line 419-422.

5.

I would suggest authors to proofread the manuscript and check the grammar in English. Especially, terminology should be correctly used. Self-identified terms will need to be fully explained at the first time.

**Author reply:**

We are very grateful to the reviewer for this comment. This manuscript was proofread.

**Specific comments:**

1.

Line161. The calculation of yields belongs to the section of Experimental Methods.

**Author reply:**

The SOA yield in Line 160-161 "Here, SOA yield was calculated as the SOA mass concentration divided by the reacted VOCs." is fixed as "Here, SOA yield was defined as the ratio of the maximum SOA mass concentration ($\mu g\ m^{-3}$) to the concentration of reacted α-pinene ($\mu g\ m^{-3}$) in the end of each experiment.", and moved to the section of Experimental method in the revised manuscript in Line 144-146.

2.

Line 200. What is the unit of decay ratio?

**Author reply:**

The unit of decay ratio is "ppb min⁻¹".

3.

Are the results corrected by the effect of dilution?

**Author reply:**

We are unsure which part of this manuscript is this this comment corresponds to.

As far as I know, there is no dilution involved in this manuscript.

4.

The quantification of the particle concentration is important in this work. Can authors show the simultaneous measurements of SMPS, and state also the collection efficiency used for AMS measurement? What is the uncertainty of the yields in Table 1?

**Author reply:**

The SMPS result and AMS result is compared based on the previous expreriments. The comparation of the SOA mass concentration observed by SMPS and the signal of AMS is shown in below. The AMS results are consistent with that of SMPS. The following figure was added in the SI.

[Figure]

The uncertainty of SOA yield was based on the system error of AMS. We added the error bars in the Fig. 1, and it is changed as below.

[Figure]

Figure 1. SOA yield from α-pinene photooxidation with different initial NOx concentrations under two levels of VOCs. The error bars were determined on the system error of AMS.

Line230. Does 'gas-particle distribution coefficients' mean gas-particle partition coefficients?

**Author reply:**

We are very grateful to the reviewer for this comment. The "gas-particle partition coefficients" is more accurate. The "gas-particle distribution coefficients" in the manuscript have been changed as "gas-particle partition coefficients".

6.

The period in each experiment for yields determination is not clear. This could be important when comparing the different SOA in batch-mode studies.

**Author reply:**

The SOA yield was calculated based on the maximum SOA mass concentration. The photooxidation time for each group of experiments is 3 h in this study. For the low NOx condition of Exp. 1, 2, 7, 8, and 9, the α-pinene was not fully consumed in the end of the photooxidation, and the SOA mass concentration still increase in the end of the photooxidation. For the other experiments, the SOA mass concentration remain constant at the end of each experiment after the SOA wall loss corrected. But it should be noted that the maximum SOA mass concentration is equal to the final SOA mass concentration in the end of each experiment.

As the response to Specific comments #1, The SOA yield was defined in Line 160-161 "Here, SOA yield was calculated as the SOA mass concentration divided by the reacted VOCs." To clarify the statement, it is fixed as "Here, SOA yield was defined as the ratio of the maximum SOA mass concentration ($\mu g\ m^{-3}$) to the concentration of reacted α-pinene ($\mu g\ m^{-3}$) in the end of each experiment.", and it was moved to the section of Experimental method.

7.

Line232. 'into the aerosol phase are larger at higher concentrations' of particle phase? The statement is not clear.

**Author reply:**

The "aerosol phase" is changed as "particulate phase".

The calculation of the two-product model (line237-248, equation 1, Tabe 2) should be moved into the section of Experimental Methods.

**Author reply:**

We are very grateful to the reviewer for this comment. The two-product model used in this study is a data analysis method. We think the calculation of the two-product model is reasonable in the section of discussion. In addition, some previous studies also described the calculation of the two-product model in the Discussion, i.e. Yang et al., (2020); Joo et al. (2019); Boyd et al. (2017); Chen et al. (2016).(Yang et al., 2020;Joo et al., 2019;Boyd et al., 2017;Chen et al., 2016)

9.

Line259-261. The sentence is not clear. The results do not show any change of the volatility. How can it be concluded that 'the volatility of semi volatility products produced through α-pinene photooxidation increased with initial NOx concentration'?

**Author reply:**

We are very grateful to the reviewer for this comment. Here, the $K_{om,2}$ ($m^3$ $\mu g^{-1}$) are the gas-particle partitioning equilibrium constants for semi-volatility products. The decreased value of $K_{om,2}$ illustrates the volatility of semi volatility products was decreased.

For the illustration more clearly, this sentence is changed as "The decreased value of $K_{om,2}$ meant that the volatility of semi-volatility products produced through α-pinene photooxidation increased with initial NOx concentration." in the revised manuscript in Line 299-301.

10.

Line266-275 repeat that yields increase with increasing NOx concentration. Please get them streamlined and delete the unnecessary repeats. Further, why the increasing

yields with increasing NOx are interpreted to be the contribution of higher contributions of semi-volatile products? Do authors mean higher NOx concentrations lead to a higher fraction of semi-volatile products among total products than at lower NOx concentrations?

**Author reply:**

We are very grateful to the reviewer for this comment.

According to the Odum two-product model the fitted curves for each experiment gradually moved to the lower position with the increase of NOx, which indicated that the volatility of photooxidation products increases with the increasing initial NOx concentration. At the same time, the higher NOx concentration enhances the RO reaction path, which also indicates the formation ability of SOA was inhibited by NOx. But in low NOx conditions, the consumption of VOCs and the SOA mass concentration was increased with initial NOx concentrations. Based on the nature of Odum model, the SOA yield was increases with increasing SOA mass concentration in each photooxidation experiment through the gradual increase in the gas-particle partition coefficients of photooxidation product. The inhibition of SOA formation by the roles of chemical processes (the branching of RO2 react with RO2/HO2 or NO) and facilitation of SOA formation by the physical processes (aerosol/gas-phase distribution) jointly effect the SOA yield. With the increase of NOx concentration, more SOA was formed. The SOA mass concentration was increased from 26.0 μg m$^{-3}$ to 54.3 μg m$^{-3}$ as the initial NOx concentration increased from 12 ppb to 68 ppb. Hence, we believe that the increased SOA yield with increasing NOx concentration is responsible by the aerosol/gas-phase partition of semi-volatile products.

In order to express more clearly, the manuscript in Line 266-275 is fixed as "Due to the lower consumption rate of VOCs and low AOC, α-pinene was not completely consumed at the end of the photooxidation period under low-NOx conditions, and the consumption of α-pinene was increased with the increasing NOx concentration. The increased VOC consumption resulted in higher concentrations of photooxidation products generated in the chamber. Consequently, when the initial NOx concentration increased from 12 ppb to 25 ppb and further to 68 ppb, the mass concentration of SOA increased from 26.0 μg m$^{-3}$ to 43.8 μg m$^{-3}$ and eventually reached 54.3 μg m$^{-3}$. Because of the positive correlation between SOA yield and SOA mass concentration, although

a gradual downward shift in the fitting curve of the two-product model was observed with increasing NOx levels, the higher SOA mass concentration still resulted in an increase in SOA yield from 6.5% to 8.0% when the initial NOx concentration increased from 12 ppb to 68 ppb." in the revised manuscript in Line 305-316.

We do not mean the higher NOx concentrations lead to a higher fraction of semi-volatile products among total products than at lower NOx concentrations. We want to express the aerosol/gas-phase partition ratio was increased with increasing formation of semi-volatile products. For clarify, The sentence of "The enhancement of the SOA yield with increasing NOx concentrations can be attributed to the increased ratio of the aerosol/gas phase partition resulting from higher concentrations of semi-volatile photooxidation products." in Line 277-280 is fixed as Hence, the enhancement of the SOA yield with increasing NOx concentrations can be attributed to the increased partition ratio of semi-volatile photooxidation products between aerosol and gas phase when more photooxidation products were formed." in the revised manuscript in Line 319-322.

11.

Line283-286. What do the 'volatile oxidation products' refer to? Oxidation products in semi volatility or in all volatility groups? Do the proportions refer to the amount of total semi-volatile oxidation product in the sum of all gas- and particle-phase oxidation products? These confuse me. Also, I would suggest authors to give citations and explain why the assumption given here is correct.

**Author reply:**

We are very grateful to the reviewer for this comment.

In the study of Chen et al. (2022), the photooxidation products are grouped into five classes based on their saturation vapor pressure (C*), i.e., volatile organic compounds (VOC), intermediate volatility organic compounds (IVOC), semivolatile organic compounds (SVOC), low volatility organic compounds (LVOC), and extremely low volatility organic compounds (ELVOC). And the "different volatile oxidation products" in our manuscript is means the "VOC, IVOC, SVOC, LVOC, and ELVOC" in all volatility groups.

To clarify the statement, we changed the sentence in line 280-286 of the revised manuscript as "Chen et al. (2022) categorized the photooxidation products into five classes based on their saturated vapor pressure (C*), and relative content of different classes of volatile products in both gas and particulate phase were compared. The contributions of semi-volatile oxidized products in the particulate phase were larger, but the proportion of semi-volatile oxidized products in gas-phase intermediate products was lower when experiments had higher VOC consumption and SOA yields. This result indicated that the proportion of semi-volatile organic products condensed into the particulate phase relative to the total formation of semi-volatile organic products was larger when more VOCs were consumed." in the revised manuscript in Line 323-331.

12.

Line367-373. The calculation belongs to the section of Experimental Methods.

**Author reply:**

We are very grateful to the reviewer for this comment. And the calculation of NOCs has moved to the section of Experimental Methods.

**Technical comments:**

1.

line60. Abbreviations should be identified when they show up for the first time.

**Author reply:**

The Abbreviations of VOCs is identified in Line 51. The identification of OH was added in Line 60.

2.

Figure6. In the legend, 'complately' is incorrect.

**Author reply:**

The "complately" is fixed as "almost".

References

Takeuchi, M., and Ng, N. L.: Chemical composition and hydrolysis of organic nitrate aerosol formed from hydroxyl and nitrate radical oxidation of α-pinene and β-pinene, Atmos. Chem. Phys., 19, 12749-12766, 10.5194/acp-19-12749-2019, 2019.

Xu, L., Suresh, S., Guo, H., Weber, R. J., and Ng, N. L.: Aerosol characterization over the southeastern United States using high-resolution aerosol mass spectrometry: spatial and seasonal variation of aerosol composition and sources with a focus on organic nitrates, Atmos. Chem. Phys., 15, 7307-7336, 10.5194/acp-15-7307-2015, 2015.

**References**

Akherati, A., Cappa, C. D., Kleeman, M. J., Docherty, K. S., Jimenez, J. L., Griffith, S. M., Dusanter, S., Stevens, P. S., and Jathar, S. H.: Simulating secondary organic aerosol in a regional air quality model using the statistical oxidation model - Part 3: Assessing the influence of semi-volatile and intermediate-volatility organic compounds and NOx, Atmos. Chem. Phys., 19, 4561-4594, 10.5194/acp-19-4561-2019, 2019.

Boyd, C. M., Nah, T., Xu, L., Berkemeier, T., and Ng, N. L.: Secondary organic aerosol (SOA) from nitrate radical oxidation of monoterpenes: Effects of temperature, dilution, and humidity on aerosol formation, mixing, and evaporation, Environ Sci Technol, 51, 7831-7841, 10.1021/acs.est.7b01460, 2017.

Chen, C. L., Kacarab, M., Tang, P., and Cocker, D. R.: SOA formation from naphthalene, 1-methylnaphthalene, and 2-methylnaphthalene photooxidation, Atmospheric Environment, 131, 424-433, 10.1016/j.atmosenv.2016.02.007, 2016.

Chen, T. Z., Zhang, P., Chu, B. W., Ma, Q. X., Ge, Y. L., Liu, J., and He, H.: Secondary organic aerosol formation from mixed volatile organic compounds: Effect of RO2 chemistry and precursor concentration, NPJ Clim. Atmos. Sci., 5, 95, 10.1038/s41612-022-00321-y, 2022.

Jiang, X., Tsona, N. T., Jia, L., Liu, S., Zhang, H., Xu, Y., and Du, L.: Secondary organic aerosol formation from photooxidation of furan: effects of NOx and humidity, Atmos. Chem. Phys., 19, 13591-13609, 10.5194/acp-19-13591-2019, 2019.

Joo, T., Rivera-Rios, J. C., Takeuchi, M., Alvarado, M. J., and Ng, N. L.: Secondary organic aerosol formation from reaction of 3-methylfuran with nitrate radicals, ACS Earth and Space Chemistry, 3, 922-934, 10.1021/acsearthspacechem.9b00068, 2019.

Kolesar, K. R., Chen, C., Johnson, D., and Cappa, C. D.: The influences of mass loading and rapid dilution of secondary organic aerosol on particle volatility, Atmos. Chem. Phys., 15, 9327-9343, 10.5194/acp-15-9327-2015, 2015.

Odum, J. R., Hoffmann, T., Bowman, F., Collins, D., Flagan, R. C., and Seinfeld, J. H.: Gas/particle

partitioning and secondary organic aerosol yields, Environ Sci Technol, 30, 2580-2585, 10.1021/es950943+, 1996.

Qi, X., Zhu, S. P., Zhu, C. Z., Hu, J., Lou, S. R., Xu, L., Dong, J. G., and Cheng, P.: Smog chamber study of the effects of NOx and NH$_3$ on the formation of secondary organic aerosols and optical properties from photo-oxidation of toluene, Sci. Total. Environ., 727, ARTN 138632, 10.1016/j.scitotenv.2020.138632, 2020.

Sarrafzadeh, M., Wildt, J., Pullinen, I., Springer, M., Kleist, E., Tillmann, R., Schmitt, S. H., Wu, C., Mentel, T. F., Zhao, D., Hastie, D. R., and Kiendler-Scharr, A.: Impact of NOx and OH on secondary organic aerosol formation from β-pinene photooxidation, Atmos. Chem. Phys., 16, 11237-11248, 10.5194/acp-16-11237-2016, 2016.

Takeuchi, M., and Ng, N. L.: Chemical composition and hydrolysis of organic nitrate aerosol formed from hydroxyl and nitrate radical oxidation of α-pinene and β-pinene, Atmos. Chem. Phys., 19, 12749-12766, 10.5194/acp-19-12749-2019, 2019.

Takeuchi, M., Berkemeier, T., Eris, G., and Ng, N. L.: Non-linear effects of secondary organic aerosol formation and properties in multi-precursor systems, Nat. Commun., 13, 10.1038/s41467-022-35546-1, 2022.

Valorso, R., Aumont, B., Camredon, M., Raventos-Duran, T., Mouchel-Vallon, C., Ng, N. L., Seinfeld, J. H., Lee-Taylor, J., and Madronich, S.: Explicit modelling of SOA formation from α-pinene photooxidation: sensitivity to vapour pressure estimation, Atmos. Chem. Phys., 11, 6895-6910, 10.5194/acp-11-6895-2011, 2011.

Xu, L., Suresh, S., Guo, H., Weber, R. J., and Ng, N. L.: Aerosol characterization over the southeastern United States using high-resolution aerosol mass spectrometry: spatial and seasonal variation of aerosol composition and sources with a focus on organic nitrates, Atmos. Chem. Phys., 15, 7307-7336, 10.5194/acp-15-7307-2015, 2015.

Yang, Z. M., Tsona, N. T., Li, J. L., Wang, S. Y., Xu, L., You, B., and Du, L.: Effects of NOx and SO$_2$ on the secondary organic aerosol formation from the photooxidation of 1,3,5-trimethylbenzene: A new source of organosulfates, Environ. Pollut., 264, 114742, 10.1016/j.envpol.2020.114742, 2020.

Zhao, D., Schmitt, S. H., Wang, M., Acir, I. H., Tillmann, R., Tan, Z., Novelli, A., Fuchs, H., Pullinen, I., Wegener, R., Rohrer, F., Wildt, J., Kiendler-Scharr, A., Wahner, A., and Mentel, T. F.: Effects of NOx and SO$_2$ on the secondary organic aerosol formation from photooxidation of α-pinene and limonene, Atmos. Chem. Phys., 18, 1611-1628, 10.5194/acp-18-1611-2018, 2018.

---

## Author Comment (AC2)

**Response to Reviewers**

Ms. Ref. No.: EGUSPHERE-2024-1599

Changes in Aerosol/Gas-Phase Distribution Ratio of Semi-Volatile Products Affect Secondary Organic Aerosol Formation with NOx from α-Pinene Photooxidation

Dear Editor:

We greatly appreciate the time and effort that the editor and reviewer spent in reviewing our manuscript. After reading the comments from the reviewers, we have carefully revised our manuscript. All the changes we made are marked in red. Our responses to the comments are itemized below. The referee's comments are in black, authors' responses are in blue.

Anything for our paper, please feel free to contact me via **ghwang@geo.ecnu.edu.cn.**

All the best

Wang Gehui

October, 2024

**Reviewer #1**

Liu et al. investigated the photooxidation of α-pinene using a smog chamber. They studied the SOA yields under different NOx concentrations with low- and high-VOC concentrations. Based on the two-product model and chemical composition measurement, semi-volatile oxidation products were suggested as the main components of the α-pinene SOA particles.

**The study falls into the scope of ACP and will be of interest to the aerosol community.** However, the use of atmospherically irrelevant VOC concentrations and the use of the language have significantly weakened the quality of the work. The manuscript will require major revisions. The comments below need to be considered and addressed before the manuscript can be considered for final publication.

**Major Comments**

The reported mass concentrations were claimed to be corrected with wall losses. Was it corrected for particle wall loss and/or vapor wall loss? However, there is no detailed information about the correction procedure. It is unclear to me, but also to general readers, how the mass concentration was corrected. Was the α-pinene corrected for vapor loss?

**Author reply:**

We are very grateful to the reviewer for this comment.

Based on our previous studies, the VOCs concentration was almost unchanged when let it stand for 5 hours in the chamber. Therefore, the wall loss of α-pinene is negligible.

All the particle mass concentration was corrected with the same way of Jiang et al. (2020) and Pathak et al. (2007) to constrained the influence of wall losses of different SOA formed with different experiment conditions. For each experiment, we continued to monitor the particle concentration in the dark condition for 1 hour, and recalculated the particle wall loss constant according to the variation of particle concentration. After the wall loss correction, the particle mass concentration was almost constant (New Fig.S1), we believe that our results are reliable and credible.

To clarify the statement, we added the sentences in line 139 of the revised manuscript as : "The particle wall loss rates were detected at the end of the chamber experiment after the UV-lamps were turned off, and the mass concentration was

corrected with the same way of Jiang et al. (2020) and Pathak et al. (2007). After the wall loss correction, the particle mass concentration was almost constant (Fig.S1), the different wall loss effect caused by gaseous oxidation products formed in the different experiment conditions have been remedied."

Did the SOA mass concentration remain constant at the end of each experiment? How was the SOA yield defined in this study? Was the maximum or final SOA mass concentration used for the SOA yield calculation?

**Author reply:**

The SOA yield was calculated based on the maximum SOA mass concentration. The photooxidation time for each group of experiments is 3 h in this study. For the low NOx condition of Exp. 1, 2, 7, 8, and 9, the α-pinene was not fully consumed in the end of the photooxidation, and the SOA mass concentration still increase in the end of the photooxidation. For the other experiments, the SOA mass concentration remain constant at the end of each experiment after the SOA wall loss corrected. But it should be noted that the maximum SOA mass concentration is equal to the final SOA mass concentration in the end of each experiment.

The SOA yield was defined in Line 160-161 "Here, SOA yield was calculated as the SOA mass concentration divided by the reacted VOCs." To clarify the statement, it is fixed as "Here, SOA yield was defined as the ratio of the maximum SOA mass concentration ($\mu g\ m^{-3}$) to the concentration of reacted α-pinene ($\mu g\ m^{-3}$) in the end of each experiment.", and it was moved to the section of Experimental method in line 144 of the revised manuscript. word 中复制的文字保存在一边 随时用

Lines 163 and 166: How were the low- and high-VOC experiments defined? Did Exp. 1 – Exp. 6 belong to low-VOC experiments? And the Exp. 7 - Exp. 14 belonged to high-VOC ones, didn't they?

**Author reply:**

As the reviewer pointed, Exp. 1 – Exp. 6 belong to low-VOC experiments, and the Exp. 7 - Exp. 14 belonged to high-VOC. To provide a better illustration of Low and

High-VOCs conditions, the following sentences have been added in the revised manuscript in line 133-136: "Two different α-pinene concentrations were used in this study. Exp. 1 to 6 were defined as low-VOC experiments, and the others were high-VOCs experiments. The concentrations of α-pinene were kept as constant as possible across low- or high-VOC experiments to ensure the effects of NOx were not obscured."

Lines 167 – 173: Even though the low-VOC experiments were chosen for analysis, the concentration of α-pinene used in the experiments is still one or two orders of magnitude higher than that in the atmosphere (Li et al., 2021). I suggest that later in the manuscript, the authors should discuss the caveat of using hundreds of ppb of α-pinene in chamber studies here and what might be different from the chemistry occurring in the ambient.

**Author reply:**

We are very grateful to the reviewer for this comment.

We add the sentences of "Though the initial concentrations of VOCs were higher than those in the atmosphere and the SOA mass loadings were vastly overrated, this study provides new insights into the nonlinear relationship of NOx with SOA yield, and may be informative to future studies with more atmospheric-relevant concentrations of reactants. Furthermore, according to the changes in the aerosol/gas-phase partition ratio of semi-volatile products with changing VOC concentrations, the proportion of semi-volatile oxidation products distributed in the gas phase would be much higher in the real atmosphere, while the ratio in particulate phase would be lower than observed in our study. Identifying of semi-volatile oxidation products in both the aerosol and gas phase will further enhance our understanding of SOA formation processes. Moreover, the higher ratio of semi-volatile oxidation products distributed in the gas phase in the atmosphere suggests a more pronounced influence of environmental factors, such as acid-catalyzed heterogeneous reactions and liquid-phase reactions, on SOA formation compared to the laboratory studies." in the end of Section 3.3.

Lines 178 – 180: How was the SOA yield in this study compared to the literature data?

**Author reply:**

We are very grateful to the reviewer for this comment.

The trend of SOA yield with initial NOx concentration is illustrated below.

[Figure]

(Aruffo et al., 2022)

(Liu et al., 2019)

(Kroll et al., 2006)

Because the experiment conditions are different among these studies, it is hard to include a figure for such a comparison. To clarify the statement, we changed the sentence in line 208-210 of the revised manuscript as "Like our study, similar trend of SOA yields first increase and then gradually decreasing with initial NOx concentrations have been widely observed in previous studies (Aruffo et al., 2022;Liu et al., 2019b;Kroll et al., 2006)"

**Author reply:**

The suppressed autooxidation of RO2 and reduction in HOMs were not observed in this study. This was referenced in previous studies. This sentence was fixed as "Additionally, numerous other studies have indicated that the autooxidation of $RO_2$ can be effectively suppressed through the $RO_2 + NO / NO_2$ reaction, which results in the reduction of HOMs formation and subsequently contributes to decreased SOA yields under high-NOx conditions (Yu et al., 2022;Laskin et al., 2018)" in the revised manuscript in line 216-219.

Lines 190 - 212: The atmospheric oxidizing capacity (AOC) describes the apparent decay rate of VOCs. AOC is not a term widely used in the atmospheric science community. To help readers understand AOC, the authors are encouraged to give detailed explanations of both the mathematical and physical meanings of AOC. In the context of the work presented here, AOC describes how OH and O3 together reacted with α-pinene under different NOx levels. Instead of using AOC here, using a simple box model to determine how much α-pinene was consumed by individual oxidants (i.e., OH and O3) under different NOx conditions would be more beneficial.

**Author reply:**

We are very grateful to the reviewer for this comment.

A substantial amount of research on AOC studies have been reported, i.e. Dai et al.(2023); Feng et al. (2021a); Feng et al. (2021b); Pawar et al. (2024); Zhao et al. (2020); Feng et al. (2019); Ma et al. (2024) (Dai et al., 2023;Feng et al., 2021a;Feng et al., 2021b;Pawar et al., 2024;Zhao et al., 2020;Feng et al., 2019;Ma et al., 2024).

VOCs in the atmosphere are removed through atmospheric oxidation reactions. The removal rate is related to the oxidizing ability of the atmosphere, and it is apparently the strength of atmospheric oxidation. AOC determines the removal rate of trace gases and also the production rates of secondary pollutants (Prinn, 2003), is the essential driving force of atmospheric chemistry in forming complex air pollution in

the troposphere and the near-surface atmosphere (Cheng et al., 2007; Lin and Zhao, 2009).

The consumption ratio of VOCs by OH was calculated by the online data of O3 and VOCs concentrations. The consumption ratio of VOCs by OH was increased with the initial NOx concentration. However, there is no clear connection between our analysis of SOA yield and consumption ratio of VOCs by OH or $O_3$. When we analyzed the combined oxidizing capacity of O3 and OH as a whole, which represents the amount of VOCs consumption is more reasonable to analyze the effect of the aerosol/gas-phase distribution ratio of the semi-volatile products on SOA yield.

In order to express the meanings of AOC more clearly, the following sentence of " The atmospheric oxidizing capacity (AOC), which indicates the oxidizing ability of the atmosphere, is significantly influenced by NOx (Wang et al., 2023)." is fixed as: "Atmospheric oxidizing capacity (AOC) is an essential driving force of the oxidizing ability of the atmosphere, which determines the removal rate of trace gases and also the production rates of secondary pollutants (Lin et al., 2009;Ma et al., 2024). It has been shown that increases in AOC are essential drivers of increases in SOA mass concentration in the troposphere (Li et al., 2023;Feng et al., 2019). The strength of AOC is significantly influenced by NOx (Wang et al., 2023)." in the revised manuscript in line 224-230.

Eq 1 and Table 2: What boundaries were set for the starting points of Kom and α? Why is the value of Kom,1 always equal to 0.19? Did the author put any constraint on the Kom,1? Uncertainties need to be provided for the fitting parameters in two product models in Table 2.

**Author reply:**

SOA yield has been described by a semi-empirical model based on the absorptive gas-particle partitioning of products (Ng et al., 2007b; Song et al., 2005). The SOA yield (Y) of an individual precursor is calculated via

$$Y = M_0 \sum_{i=1}^{n} \left( \frac{\alpha_i K_{om,i}}{1 + K_{om,i} M_0} \right)$$

Previous studies on SOA yields from both biogenic and anthropogenic precursors suggested that a two-product model (n = 2) can accurately and adequately describe the experimental data with the model parameters $a_1$, $a_2$, $K_{om,1}$, and $K_{om,2}$. We assumed that similar low-volatility species were generated in each NOx scenarios, as the same set in the previous study (Li et al., 2016a; Li et al., 2016b). The lower-volatility partitioning parameter ($K_{om,1}$) in all yield curve fitting are assigned to a fixed value by assuming similar lower-volatility compounds are formed during all photooxidation experiments. The experimental fitting parameters in the two-product model were determined by minimizing the sum of the squared of the residual. Each experimental yield data can be fitted well by the two-product model.

To illustrate more clearly, the following sentences have been added in the revised manuscript in line 278-283: "We assumed that similar low-volatility species were generated in each NOx scenarios, as the same set in the previous study (Yang et al., 2020;Li et al., 2016). The lower-volatility partitioning parameter ($K_{om,1}$) was assigned a fixed value in all yield curve fittings based on the assumption that similar lower-volatility compounds are formed during all photooxidation experiments."

As far as we know, no uncertainties for the fitting parameters were reported. But we also think the uncertainties is very important. In order to better display the reliability of fitting results, the correlations between the models and experiments data were added in Table 2.

Table 2 Parameters of the two product model for α-derived SOA under different initial NOx concentration.

| Initial NOx conc. (ppb) | $\alpha_1$ | $K_{om,1}$ ($m^3\ \mu g^{-1}$) | $\alpha_2$ | $K_{om,2}$ ($m^3\ \mu g^{-1}$) | $R^2$ |
|---|---|---|---|---|---|
| 12 | 0.048 | 0.19 | 0.28 | 0.0040 | 0.9991 |
| 25 | 0.038 | 0.19 | 0.30 | 0.0039 | 0.9997 |
| 68 | 0.028 | 0.19 | 0.32 | 0.0037 | 0.9989 |
| 150 | 0.019 | 0.19 | 0.33 | 0.0031 | 0.9992 |
| 337 | 0.017 | 0.19 | 0.35 | 0.0019 | 0.9990 |
| 600 | 0.014 | 0.19 | 0.38 | 0.0016 | 0.9987 |

Lines 283 – 286: The assumption for the constant proportions of different volatile oxidation products is out of sense. Variations in α1/α2 ratios with initial NOx concentration have been mentioned in the part associated with the two-product model.

**Author reply:**

This assumption is not for our study, but for the result in the study of Chen et al. (2022). The variations in α1/α2 ratios with initial NOx concentration is used to evaluate the distribution of different volatile oxidation products between gas and aerosol-phase. But not for the ratio of the photooxidation products with different vapor. We are very grateful to the reviewer for this comment. The assumption was deleted from the manuscript. In addition, to clarify the statement, we changed the sentence in line 323-331 in revised manuscript as "Chen et al. (2022) categorized the photooxidation products into five classes based on their saturated vapor pressure ($C^*$), and relative content of different classes of volatile products in both gas and particulate phase were compared. The contributions of semi-volatile oxidized products in the particulate phase were larger, but the proportion of semi-volatile oxidized products in gas-phase intermediate products was lower when experiments had higher VOC consumption and SOA yields. This result indicated that the proportion of semi-volatile organic products condensed into the particulate phase relative to the total formation of semi-volatile organic products was larger when more VOCs were consumed."

**Minor Comments**

Line 33: Unless more contexts are provided, it is unclear why "low- and high-volatility" are used here.

**Author reply:**

As the reviewer pointed, we give a description of "low- and high-volatility" in the revised manuscript in line 133 as "Two different α-pinene concentrations were used in this study. Exp. 1 to 6 were defined as low-VOC experiments, and the others were high-VOCs experiments."

Lines 36 – 38:" … and the change in the aerosol/gas… with increasing NOx" is too long to read and understand. Please rephrase the sentence.

**Author reply:**

This sentence is change to "The enhanced SOA yields with increasing NOx were primarily attributed to the change in the aerosol/gas-phase partition ratio, resulting from the increased formation of α-pinene photooxidation products." in the revised manuscript in line 36-38.

Lines 77 – 78: What are the other experimental conditions? Please provide examples.

**Author reply:**

The experimental conditions include oxidation conditions,NOx concentration,RH,temperature and so on. For instance, in the research conducted by Takeuchi et al., the typical seed number and volume concentrations were maintaining consistency. The initially ratio of $N_2O_5$ to α-pinene was 4. The chamber was conditioned to 5 °C and low humidity (RH < 5%).

To avoid misunderstandings, the sentence of "Based on the semi-volatile partitioning theory in SOA formation, it has been established that SOA yield is a function of SOA mass concentration when other experimental conditions are held constant (Odum et al., 1996;Takeuchi et al., 2022)." is changed as "Based on the semi-volatile partitioning theory in SOA formation, SOA yield is strongly dependent on the SOA mass concentration present in the system ($M_0$) (Odum et al., 1996;Takeuchi et al., 2022)." in the revised manuscript in line 76-79.

Line 80: "the SOA yield is often discussed as a constant" lacks clarity.

**Author reply:**

We are very grateful to the reviewer for this comment.

The sentence of "However, the SOA yield is often discussed as a constant, and the nonlinear relationships between SOA yield and initial NOx concentration reported in chamber studies do not account for the consumption of VOCs." is fixed as "However, the nonlinear relationships between SOA yield and initial NOx concentration reported in chamber studies do not account for the consumption of VOCs" in the revised manuscript in line 80-82.

Lines 92 -94: Please provide more descriptions of chemical processes and physical processes.

**Author reply:**

We are very grateful to the reviewer for this comment.

The sentence of "The roles of chemical processes are often considered due to the impacts of NOx on SOA yields, but physical processes in SOA formation are equally significant and should be given more attention." is fixed as "The roles of chemical processes (the branching of $RO_2$ reacts with $RO_2/HO_2$ or NO) are often considered due to their impacts of NOx on SOA yields, but physical processes (aerosol/gas-phase partition) in SOA formation are equally significant and should be given more attention." in the revised manuscript in line 91-94.

Lines 110 – 111: What is the volume of the chamber? What was the experimental temperature?

**Author reply:**

The volume of the chamber is 5 $m^3$. The sentence in Line 120 have shown that 5 m3 zero air was added into the chamber. For clarify, the volume of the chamber was added in the revised manuscript in Line 110-111 as "A series of α-pinene photooxidation experiments initiated by NOx were performed in a temperature controlled 5 $m^3$ photooxidation chamber."

We added the sentence of "The temperature during the photooxidation process was $25 \pm 3$ °C." in the revised manuscript in Line 128.

Line 128: What were the sources of OH and O3? Did the authors inject H2O2?

**Author reply:**

No $H_2O_2$ was added into the chamber.

Both of the O3 and OH are the coexisting oxidants in the photooxidation (Sarrafzadeh et al., 2016;Liu et al., 2017;Wang et al., 2017) . The $O_3$ are formed from the photooxidation of $NO_2$, and OH was formed from the photolysis of $O_3$ ($O_3+hv \rightarrow O(_1D)+O_2$, $O(_1D)+H_2O \rightarrow OH+OH$) and the recycle of $HO_2$ radicals (NO + $HO_2 \rightarrow NO_2 + OH$) .

For clarify, the references of Liu et al. (2017); Wang et al. (2017); Sarrafzadeh et

al. (2016) were added in the revised manuscript in Line 130.

Was the chamber operated in a batch mode or continuous mode? How long did one experiment last?

**Author reply:**

The chamber was operated in a batch mode. Each experiment last 180 min as shown in Fig.2.

We added the sentences of "The photooxidation was operated in a batch mode, and each experiment lasted 180 min." in the revised manuscript in Line 131-132.

What was the background level of NOx before any experiment?

**Author reply:**

Normally, the background level of NOx before each experiment is lower than the detect limitation (1 ppb) of NOx analyzer.

We added the background level of NOx in the revised manuscript in Line 118-120 as "This filling-purging cycle was repeated 5 times between experiments to ensure the residual particulate, α-pinene and NOx concentrations were less than 5 cm$^{-3}$, 0.5 ppb, and 1 ppb, respectively."

Figure 1: Please include error bars for the SOA yield. In addition, using different marker shapes is redundant.

**Author reply:**

We are very grateful to the reviewer for this comment.

The error bars were added, and the Fig. 1 is fixed as below.

[Figure]

Figure 1. SOA yield from α-pinene photooxidation with different initial NOx concentrations under two levels of VOCs. The error bars were determined on the system error of AMS.

Lines 231 and 232: What concentrations? Please clarify it.

**Author reply:**

This sentence is fixed as "the distribution coefficients of semi-volatile substances into the particulate phase are larger the higher their concentrations are" in the revised manuscript in Line 266.

Lines 337 -341: Falling into the lower area of SV-OOA does not necessarily mean that the semi-volatile products were the main components of α-pinene SOA (Paciga et al., 2016; Kang et al., 2022).

**Author reply:**

We are very grateful to the reviewer for this comment.

Both of the study of Paciga et al. and Kang et al. assess the volatility of SOA based on the $C^*$ value. However, the $C^*$ cannot be obtained from our current experimental conditions.

This method for the assessment of organic aerosol volatility based on $F_{43}$ *vs*. $F_{44}$

has been used in many studies, i.e. (Ng et al., 2010;Hao et al., 2014;Reyes-Villegas et al., 2016;Singh et al., 2019), and these references are added into the manuscript.

For clarify, the sentence of "The AMS results provided direct evidence that semi-volatile products were the main components of α-pinene SOA formed through NOx photooxidation." is fixed as "The AMS results suggested that the α-pinene SOA formed through NOx photooxidation exhibited semi-volatile characteristics." in the revised manuscript in Line 385-386.

Lines 359 – 377: How would the method of estimating NOC used here differ from that of Kiendler‑Scharr et al. (2016)?

**Author reply:**

Our study estimated the concentration of $NO_2^+$ fragmentated from the organic nitrate, and calculated $NO^+$ from the organic nitrate based on the $R_{ON}$. In the study of Kiendler-Scharr et al. (2016), they first determine the fraction of particulate organic nitrate ($pOrgNO_{3frac}$) in the measured total nitrate, and then calculated the mass concentration of organic nitrate by multiplying the measured total nitrate ($NO_{3total}$) with the fraction of $pOrgNO_3$

The $R_{AN}$, $R_{ON}$, and $R_{means}$ in our studies is expressed as $pInNO_3$, $pOrgNO_3$, and $R_{measured}$ in the study of Kiendler‑Scharr et al. (2016), respectively. In the review of Ng et al. (2017), both methods were mentioned simultaneously for the calculation of NOCs. The method used in our study and Kiendler‑Scharr et al. (2016) for the estimation of NOC through AMS is in agreement with each other.

**Technical Comments**
Line 26: "Atmospheric α-pinene" sounds very odd. Please just use "α-pinene".

**Author reply:**

Fixed.

In many places, I found there are two terms for nitrogen oxides, i.e., NOx and NOx.

Please make it consistent throughout the manuscript.

**Author reply:**

We are very grateful to the reviewer for this comment, and all the "NOₓ" is fixed as "NOx"

Line 50: Wang et al., 2016 investigated the mechanism behind the sulfate formation. I don't know why this paper is cited here.

**Author reply:**

We are very grateful to the reviewer for this comment.

This reference is deleted, and the references of Wei et al. (2021); Matsui and Liu (2022) were added here.

Line 52: Lv et al., 2022 is about the gas-to-particle partitioning of WSOC. This paper seems irrelevant to be cited here.

**Author reply:**

We are very grateful to the reviewer for this comment. This reference is deleted.

Line 54: It should be "many" instead of "much".

**Author reply:**

We are very grateful to the reviewer for this comment. We think "much" is more reasonable here. We fixed this sentence to "much research is still needed to fully understand the formation mechanisms of SOA" to make the language clearer.

Line 141: What was the E/N value used for the PTR? How long was the sampling line of PTR connected to the chamber outlet?

**Author reply:**

The E/N is z135 Td. The sampling line of PTR connected to the chamber outlet is

1.2 m.

The operation of PTR-MS is fixed as: "The drift tube of the PTR-tof-MS was operated at 60.0 °C ($T_{drift}$), 2.30 mbar ($P_{drift}$), and 600V ($U_{drift}$), which resulted in an $E/N$ value of 135 Td."

Line 151: Please provide more information about the scanning mobility particle sizer.

**Author reply:**

The SMPS information is added as: "The SOA mass concentrations obtained from AMS measurements were compared and corrected through a scanning mobility particle sizer (SMPS, TSI Inc., USA). The sheath and aerosol flow rate used in the SMPS were set to 3 and 0.3 L min$^{-1}$, respectively. Each scan lasted 240 s and the scanning range of aerodynamic equivalent diameter of SOA was 13.8–749.9 nm. For the calculation of SOA mass concentration in this study, an assumed density of 1.2 g cm$^{-3}$ for α-pinene SOA was taken into account (Aruffo et al., 2022)." in the revised manuscript in Line 163-169.

Line 164: Is the word "defend" a typo?

**Author reply:**

Fixed as "defined"

Figure 3: What do the lines' colors stand for? It is very hard to follow the order of the markers without carefully reading the legend.

**Author reply:**

Fig. 3 is fixed as below.

[Figure]

Figure 3. SOA yields as a function of organic aerosol mass concentration $M_0$ of α-pinene at different initial NOx concentrations. The simulated SOA yields based on the two-product model are shown by the solid lines.

Line 406: Is it supposed to be "…under low-VOC conditions was only 54.3 µg m-3…"?

**Author reply:**

We are very grateful to the reviewer for this comment and fixed.

Line 430: Is it VOC ratio or VOC/NOx ratio?

**Author reply:**

The VOCs ratio is correct.

For clarify, we fixed this sentence as "The ratio of SOA yield from high-VOC experiments was about 3–8 times higher than that from the low-VOC experiments, which surpassed the VOC ratio (~2.2 times) between different low- and high-VOC conditions." in the revised manuscript in Line 460-462.

References

Kang, H. G., Kim, Y., Collier, S., Zhang, Q., and Kim, H.: Volatility of springtime ambient organic aerosol derived with thermodenuder aerosol mass spectrometry in seoul, korea, Environmental Pollution, 304, 119203, 2022.

Kiendler‑Scharr, A., Mensah, A. A., Friese, E., Topping, D., Nemitz, E., Prévôt, A. S., Äijälä, M., Allan, J., Canonaco, F., and Canagaratna, M.: Ubiquity of organic nitrates from nighttime chemistry in the european submicron aerosol, Geophysical Research Letters, 43, 7735-7744, 2016.

Li, H., Canagaratna, M. R., Riva, M., Rantala, P., Zhang, Y., Thomas, S., Heikkinen, L., Flaud, P.-M., Villenave, E., and Perraudin, E.: Atmospheric organic vapors in two european pine forests measured by a vocus ptr-tof: Insights into monoterpene and sesquiterpene oxidation processes, Atmos Chem Phys, 21, 4123-4147, 2021.

Paciga, A., Karnezi, E., Kostenidou, E., Hildebrandt, L., Psichoudaki, M., Engelhart, G. J., Lee, B.-H., Crippa, M., Prévôt, A. S., and Baltensperger, U.: Volatility of organic aerosol and its components in the megacity of paris, Atmos Chem Phys, 16, 2013-2023, 2016.

**References**

Aruffo, E., Wang, J., Ye, J., Ohno, P., Qin, Y., Stewart, M., McKinney, K., Di Carlo, P., and Martin, S. T.: Partitioning of organonitrates in the production of secondary organic aerosols from α-pinene photo-oxidation, Environ Sci Technol, 56, 5421-5429, 10.1021/acs.est.1c08380, 2022.

Chen, T. Z., Zhang, P., Chu, B. W., Ma, Q. X., Ge, Y. L., Liu, J., and He, H.: Secondary organic aerosol formation from mixed volatile organic compounds: Effect of RO2 chemistry and precursor concentration, NPJ Clim. Atmos. Sci., 5, 95, 10.1038/s41612-022-00321-y, 2022.

Dai, J. N., Brasseur, G. P., Vrekoussis, M., Kanakidou, M., Qu, K., Zhang, Y. J., Zhang, H. L., and Wang, T.: The atmospheric oxidizing capacity in China - Part 1: Roles of different photochemical processes, Atmos. Chem. Phys., 23, 14127-14158, 10.5194/acp-23-14127-2023, 2023.

Feng, T., Zhao, S. Y., Bei, N. F., Wu, J. R., Liu, S. X., Li, X., Liu, L., Qian, Y., Yang, Q. C., Wang, Y. C., Zhou, W. J., Cao, J. J., and Li, G. H.: Secondary organic aerosol enhanced by increasing atmospheric oxidizing capacity in Beijing-Tianjin-Hebei (BTH), China, Atmos. Chem. Phys., 19, 7429-7443, 10.5194/acp-19-7429-2019, 2019.

Feng, T., Bei, N. F., Zhao, S. Y., Wu, J. R., Liu, S. X., Li, X., Liu, L., Wang, R. N., Zhang, X., Tie, X. X., and Li, G. H.: Nitrate debuts as a dominant contributor to particulate pollution in Beijing: Roles of enhanced atmospheric oxidizing capacity and decreased sulfur dioxide emission, Atmospheric Environment, 244, 10.1016/j.atmosenv.2020.117995, 2021a.

Feng, T., Zhao, S. Y., Hu, B., Bei, N. F., Zhang, X., Wu, J. R., Li, X., Liu, L., Wang, R. N., Tie, X. X., and Li, G. H.: Assessment of atmospheric oxidizing capacity over the Beijing-Tianjin-Hebei

(BTH) area, China, J. Geophys. Res.-Atmos., 126, 10.1029/2020jd033834, 2021b.

Hao, L. Q., Kortelainen, A., Romakkaniemi, S., Portin, H., Jaatinen, A., Leskinen, A., Komppula, M., Miettinen, P., Sueper, D., Pajunoja, A., Smith, J. N., Lehtinen, K. E. J., Worsnop, D. R., Laaksonen, A., and Virtanen, A.: Atmospheric submicron aerosol composition and particulate organic nitrate formation in a boreal forestland–urban mixed region, Atmos. Chem. Phys., 14, 13483-13495, 10.5194/acp-14-13483-2014, 2014.

Jiang, X. T., Lv, C., You, B., Liu, Z. Y., Wang, X. F., and Du, L.: Joint impact of atmospheric $SO_2$ and $NH_3$ on the formation of nanoparticles from photo-oxidation of a typical biomass burning compound, Environ. Sci.-Nano, 7, 2532-2545, 10.1039/d0en00520g, 2020.

Kroll, J. H., Ng, N. L., Murphy, S. M., Flagan, R. C., and Seinfeld, J. H.: Secondary organic aerosol formation from isoprene photooxidation, Environ Sci Technol, 40, 1869-1877, 10.1021/es0524301, 2006.

Laskin, J., Laskin, A., and Nizkorodov, S. A.: Mass spectrometry analysis in atmospheric chemistry, Anal. Chem., 90, 166-189, 10.1021/acs.analchem.7b04249, 2018.

Li, L., Tang, P., Nakao, S., Chen, C. L., and Cocker Iii, D. R.: Role of methyl group number on SOA formation from monocyclic aromatic hydrocarbons photooxidation under low-NOx conditions, Atmos. Chem. Phys., 16, 2255-2272, 10.5194/acp-16-2255-2016, 2016.

Liu, S. J., Jia, L., Xu, Y., Tsona, N. T., Ge, S., and Du, L.: Photooxidation of cyclohexene in the presence of SO2: SOA yield and chemical composition, Atmos. Chem. Phys., 17, 13329-13343, 10.5194/acp-17-13329-2017, 2017.

Liu, S. J., Jiang, X. T., Tsona, N. T., Lv, C., and Du, L.: Effects of NOx, SO2 and RH on the SOA formation from cyclohexene photooxidation, Chemosphere, 216, 794-804, 10.1016/j.chemosphere.2018.10.180, 2019.

Ma, Q. X., Chu, B. W., and He, H.: Revealing the contribution of interfacial processes to atmospheric oxidizing capacity in haze chemistry, Environ Sci Technol, 58, 6071-6076, 10.1021/acs.est.3c08698, 2024.

Matsui, H., and Liu, M. X.: Substantial uncertainties in arctic aerosol simulations by microphysical processes within the global climate-aerosol model CAM-ATRAS, J. Geophys. Res.-Atmos., 127, 10.1029/2022jd036943, 2022.

Ng, N. L., Canagaratna, M. R., Zhang, Q., Jimenez, J. L., Tian, J., Ulbrich, I. M., Kroll, J. H., Docherty, K. S., Chhabra, P. S., Bahreini, R., Murphy, S. M., Seinfeld, J. H., Hildebrandt, L., Donahue, N. M., DeCarlo, P. F., Lanz, V. A., Prévôt, A. S. H., Dinar, E., Rudich, Y., and Worsnop, D. R.: Organic aerosol components observed in Northern Hemispheric datasets from Aerosol Mass Spectrometry, Atmos. Chem. Phys., 10, 4625-4641, 10.5194/acp-10-4625-2010, 2010.

Ng, N. L., Brown, S. S., Archibald, A. T., Atlas, E., Cohen, R. C., Crowley, J. N., Day, D. A., Donahue, N. M., Fry, J. L., Fuchs, H., Griffin, R. J., Guzman, M. I., Herrmann, H., Hodzic, A., Iinuma, Y., Jimenez, J. L., Kiendler-Scharr, A., Lee, B. H., Luecken, D. J., Mao, J., McLaren, R., Mutzel, A., Osthoff, H. D., Ouyang, B., Picquet-Varrault, B., Platt, U., Pye, H. O. T., Rudich, Y., Schwantes, R. H., Shiraiwa, M., Stutz, J., Thornton, J. A., Tilgner, A., Williams, B. J., and Zaveri, R. A.: Nitrate radicals and biogenic volatile organic compounds: oxidation, mechanisms, and organic aerosol, Atmos. Chem. Phys., 17, 2103-2162, 10.5194/acp-17-2103-2017, 2017.

Pathak, R. K., Stanier, C. O., Donahue, N. M., and Pandis, S. N.: Ozonolysis of alpha-pinene at atmospherically relevant concentrations: Temperature dependence of aerosol mass fractions (yields), J. Geophys. Res.-Atmos., 112, Artn D03201, 10.1029/2006jd007436, 2007.

Pawar, P. V., Mahajan, A. S., and Ghude, S. D.: HONO chemistry and its impact on the atmospheric oxidizing capacity over the Indo-Gangetic Plain, Sci. Total. Environ., 947, 10.1016/j.scitotenv.2024.174604, 2024.

Reyes-Villegas, E., Green, D. C., Priestman, M., Canonaco, F., Coe, H., Prévôt, A. S. H., and Allan,

J. D.: Organic aerosol source apportionment in London 2013 with ME-2: exploring the solution space with annual and seasonal analysis, Atmos. Chem. Phys., 16, 15545-15559, 10.5194/acp-16-15545-2016, 2016.

Sarrafzadeh, M., Wildt, J., Pullinen, I., Springer, M., Kleist, E., Tillmann, R., Schmitt, S. H., Wu, C., Mentel, T. F., Zhao, D., Hastie, D. R., and Kiendler-Scharr, A.: Impact of NOx and OH on secondary organic aerosol formation from β-pinene photooxidation, Atmos. Chem. Phys., 16, 11237-11248, 10.5194/acp-16-11237-2016, 2016.

Singh, A., Satish, R. V., and Rastogi, N.: Characteristics and sources of fine organic aerosol over a big semi-arid urban city of western India using HR-ToF-AMS, Atmos. Environ., 208, 103-112, 10.1016/j.atmosenv.2019.04.009, 2019.

Wang, T., Xue, L. K., Brimblecombe, P., Lam, Y. F., Li, L., and Zhang, L.: Ozone pollution in China: A review of concentrations, meteorological influences, chemical precursors, and effects, Sci. Total. Environ., 575, 1582-1596, 10.1016/j.scitotenv.2016.10.081, 2017.

Wei, J., Li, Z. Q., Lyapustin, A., Sun, L., Peng, Y. R., Xue, W. H., Su, T. N., and Cribb, M.: Reconstructing 1-km-resolution high-quality $PM_{2.5}$ data records from 2000 to 2018 in China: spatiotemporal variations and policy implications, Remote Sens. Environ., 252, 10.1016/j.rse.2020.112136, 2021.

Yang, Z. M., Tsona, N. T., Li, J. L., Wang, S. Y., Xu, L., You, B., and Du, L.: Effects of NOx and $SO_2$ on the secondary organic aerosol formation from the photooxidation of 1,3,5-trimethylbenzene: A new source of organosulfates, Environ. Pollut., 264, 114742, 10.1016/j.envpol.2020.114742, 2020.

Yu, S. S., Jia, L., Xu, Y. F., and Pan, Y. P.: Molecular composition of secondary organic aerosol from styrene under different NOx and humidity conditions, Atmos Res, 266, ARTN 105950, 10.1016/j.atmosres.2021.105950, 2022.

Zhao, D. D., Liu, G. J., Xin, J. Y., Quan, J. N., Wang, Y. S., Wang, X., Dai, L. D., Gao, W. K., Tang, G. Q., Hu, B., Ma, Y. X., Wu, X. Y., Wang, L. L., Liu, Z. R., and Wu, F. K.: Haze pollution under a high atmospheric oxidization capacity in summer in Beijing: insights into formation mechanism of atmospheric physicochemical processes, Atmos. Chem. Phys., 20, 4575-4592, 10.5194/acp-20-4575-2020, 2020.